# Scalable modular design of solid oxide fuel cell systems for enhanced large-scale power generation

Xinyi Wei [1,2] ✉, Arthur Waeber[1], Shivom Sharma[1], Hangyu Yu [2], Jan Van herle[2] & François Maréchal [1]

The increasing demand for renewable energy integration and scalable power generation highlights the need for efficient and cost-effective solid oxide fuel cell systems. In this study, we present a modular hybrid design framework that enables flexible solid oxide fuel cell scale-up by interconnecting standardized component modules. We introduce a series-parallel configuration that strategically leverages anode and cathode off-gas recirculation to enhance both electrical and thermal efficiency. Through a detailed case study, we demonstrate that the hybrid design achieves 66.3% electrical efficiency while reducing external water use by 59.9% and fresh air demand by 22%, outperforming conventional system designs. We further conducted a techno-economic analysis across four scale-up strategies and found that the hybrid design delivers the lowest levelized cost of electricity at 0.155 $/kWh. Through this work, we have highlighted the critical trade-offs between centralization and decentralization, high- and low-technology readiness level technologies, and economies of scale versus manufacturing capacity. We believe our findings underscore the potential of modular and standardized systems to provide scalable, efficient, and economically viable solutions for future low-carbon energy infrastructures.

The transition to renewable energy is essential for global efforts to reduce greenhouse gas emissions and establish sustainable energy systems. Climate-induced water shortages increasingly threaten power generation, underscoring the need for technologies that are less dependent on water resources[1]. Wind, solar, nuclear, and bioenergy are particularly notable for their capacity to reduce emissions of electricity, heat, and liquid/gaseous fuels production, with the potential to achieve up to 68% emission reduction[2]. Additionally, renewable energy sources exhibit over 40% lower life-cycle emissions compared to fossil fuel-based technologies integrated with carbon capture and sequestration[3,4]. Although renewable energy substantially reduces emissions compared to conventional power systems, its intermittent nature poses operational challenges. Addressing these challenges requires geopolitical stability and governance support[5], robust

planning, considerable investments[6], and advanced technologies to ensure a stable energy supply. By the end of the century, substantial investment will be needed to adapt energy infrastructure, particularly in developing regions[7].

Technological advancements, particularly in energy storage, are critical for managing the intermittency and variability of renewable energy production while ensuring energy security and grid stability[8]. With the integration of energy storage, the installation of 60 GW of renewable capacity could reduce $CO_2$ emissions by 72–90%, while curtailment could decrease up to 30%[9]. Moreover, energy storage plays a vital role in supporting climate goals, enabling renewable energy to contribute up to 66% of final energy consumption by 2050 with limiting warming to 1.5 °C[10]. Short- and long-term storage solutions complement each other in addressing mismatches between

[1]Industrial Process and Energy Systems Engineering-École Polytechnique Fédérale de Lausanne (EPFL), Sion, Valais, Switzerland. [2]Group of Energy Materials-École Polytechnique Fédérale de Lausanne (EPFL), Sion, Valais, Switzerland. ✉e-mail: xinyi.wei@epfl.ch; xinyiwei.epfl@gmail.com

energy supply and demand. Batteries are well-suited for short-term storage, while power-to-X-to-power (PtXtP) systems, such as hydrogen, provide an economically viable option for cross-seasonal storage needs[11]. Finally, integrating storage into community energy systems can reduce design costs by up to a factor of four and achieve over 80% energy self-sufficiency, significantly decreasing reliance on external energy sources[12].

While PtXtP systems are essential for enabling renewable energy integration and long-term energy storage, this study focuses on their downstream stage, X-to-power, which is critical for improving overall system efficiency and ensuring the viability of large-scale renewable deployment. Solid oxide fuel cells (SOFCs), which operate at high temperatures, have demonstrated exceptional reliability, the acceptable efficiency degradation rate of 0.2%/kh at constant power[13–16], and rapid reversibility to electrolyzer mode, making them ideal for continuous electricity production within X-to-power systems and the cogeneration of high-quality heat[17]. Moreover, due to the fuel flexibility of SOFCs, using biomethane for electricity generation can reduce emissions by up to 80% compared to non-renewable electricity production routes[4,18]. This highlights the strong potential of SOFCs for integration into future large-scale energy systems while maintaining sustainability[19].

Despite the promising prospects of SOFC systems for electricity production, they are still emerging technologies with significant scope for enhancements to broaden their large-scale application, including both system performance, particularly efficiency improvement, and cost reduction for industrial deployment[20]. From a system improvement perspective, current commercial SOFC systems generally operate without any off-gas recirculation, which indeed simplifies the system layout and control. However, the absence of recirculation results in a substantial demand for external water, which can become costly when accounting for water purification and evaporation energy requirements. A commonly adopted improvement is anode off-gas (AOG) recirculation, which utilizes the steam content in the SOFC's exhaust to reduce external water consumption and enhance overall fuel utilization. Nevertheless, the benefits of AOG recirculation do not always outweigh the additional power required to drive the recirculation blower, potentially lowering net electrical efficiency. Furthermore, since commercially available blowers cannot typically handle high-temperature gas streams, heat exchangers are required to cool and reheat the AOG flow, leading to thermal losses and increased system complexity.

While improving conversion efficiency is critical for scaling up SOFC systems[21], adopting a modular approach is equally essential for reducing system costs. Large-scale deployments generally correlate with decreased costs and reduced energy expenditures[22], fueled by market growth and increased system production rates[23]. Hence, effective strategies must be developed and implemented to ensure that SOFC technology becomes cost-effective and scalable. Currently, SOFC-based power generation systems are highly customized, limiting scalability and increasing manufacturing complexity. To overcome this, modularity should extend beyond the stack to encompass the entire system layout[13]. A standardized modular design could fix system configurations[24], enabling manufacturing plants to streamline production by focusing on fixed component modules with predefined pipeline connections[25,26]. For example, Bloom Energy has successfully deployed a 20 MW fuel cell system by interconnecting multiple 100 kW SOFC modules. However, there remains a significant gap in both industrial and academic domains regarding references, guidelines, and strategies for integrating small SOFC modules into larger-scale systems.

Thus, in this study, we focus on enhancing system efficiency while adhering to industrial constraints and propose a modular design approach for SOFC-based power generation systems. The aim is to identify scalable solutions by examining how different modules can be interconnected, either in parallel, series, or a combination of both, to form a complete system[27]. Additionally, the study evaluates which components should be centralized and which should remain decentralized, considering both system performance and economic feasibility. This ensures a balance between minimizing plant complexity and aligning with the market availability of component modules. By addressing these questions, we aim to provide a methodological framework for scaling up SOFC-based power generation systems using modular components, supporting cost-effective, market-ready solutions that are also relevant to long-term and large-scale energy storage strategies.

## Results

### SOFC hybrid design−AOG recirculation heat cascade
In this subsection, we present a hybrid design for SOFC system scale-up based on a single-layer stack module with an active area of approximately 300–320 cm² (0.1 kW) rather than a multi-layered stack to illustrate the system design concept and standardized flow information for ease of scale-up. Although such small stacks are rarely used in practice, they serve as a theoretical basis for conceptual development. Figure 1a illustrates the first potential/conventional design for effectively utilizing AOG, featuring three main component modules: an external reformer, a single-layer stack, and heat exchangers. It is crucial to clarify that this figure represents a theoretical process flow diagram; practically, systems are not operated with a single-layer stack, and reformers of such a small scale are generally not available commercially. Nevertheless, the results demonstrated here can easily be used by the industry to linearly scale up to meet specific customer requirements regarding system capacity and market availability.

In this design, fuel is heated to the required temperature for external reforming to produce syngas, which is subsequently heated to the necessary temperature for entry into the stack, where the chemical energy of the fuel is converted into electricity. The electrochemical reaction within the stack is exothermic, typically resulting in a temperature increase of 50–100 °C at the stack's outlet. For instance, Fig. 1a depicts a stack inlet temperature of 680 °C and an outlet temperature of 750 °C, with a single-pass fuel utilization of less than 0.85 (as introduced in Section "Industrial constraints and design targets"), indicating the presence of unconverted fuel downstream of the single-layer stack. This unconverted fuel is often recirculated to enhance global fuel utilization. The high-temperature AOG flow, which has enough steam, can be divided into two streams for effective utilization of available waste heat and to avoid the need for external water. One stream is mixed with the fuel before entering the reformer, eliminating the need for a heater ($Q_1 = 0$ kW) before the reformer and potentially negating the duty required by the adiabatic reformer ($Q_R = 0$ kW) if the flow is sufficiently large. The other AOG stream is mixed with the output from the reformer, potentially removing the need for a heater ($Q_2 = 0$ kW) before the stack. Figure 1a shows the standardized molar flow rates, indicating that the required AOG flows by the reformer and single-layer stack are approximately three and ten times greater than that of the fresh fuel.

The primary challenge in the conventional design in Fig. 1a is overcoming the pressure losses in the reformer and the stack to facilitate upstream mixing of AOG flow. This is typically addressed by employing an AOG blower, though high-temperature blower efficiencies are low. As a substitute, the low-temperature blower can be used, but this adds complexity due to the requirement of additional coolers/heaters, thereby increasing the number of pipeline connections. Alternatively, two ejectors could be used to manage the pressure differential, though this approach is uncommon and too customized, and may still require further research before practical industry implementation.

In this study, we have proposed an approach for scaling up SOFC systems, referred to as the hybrid design, as presented in Fig. 1b.

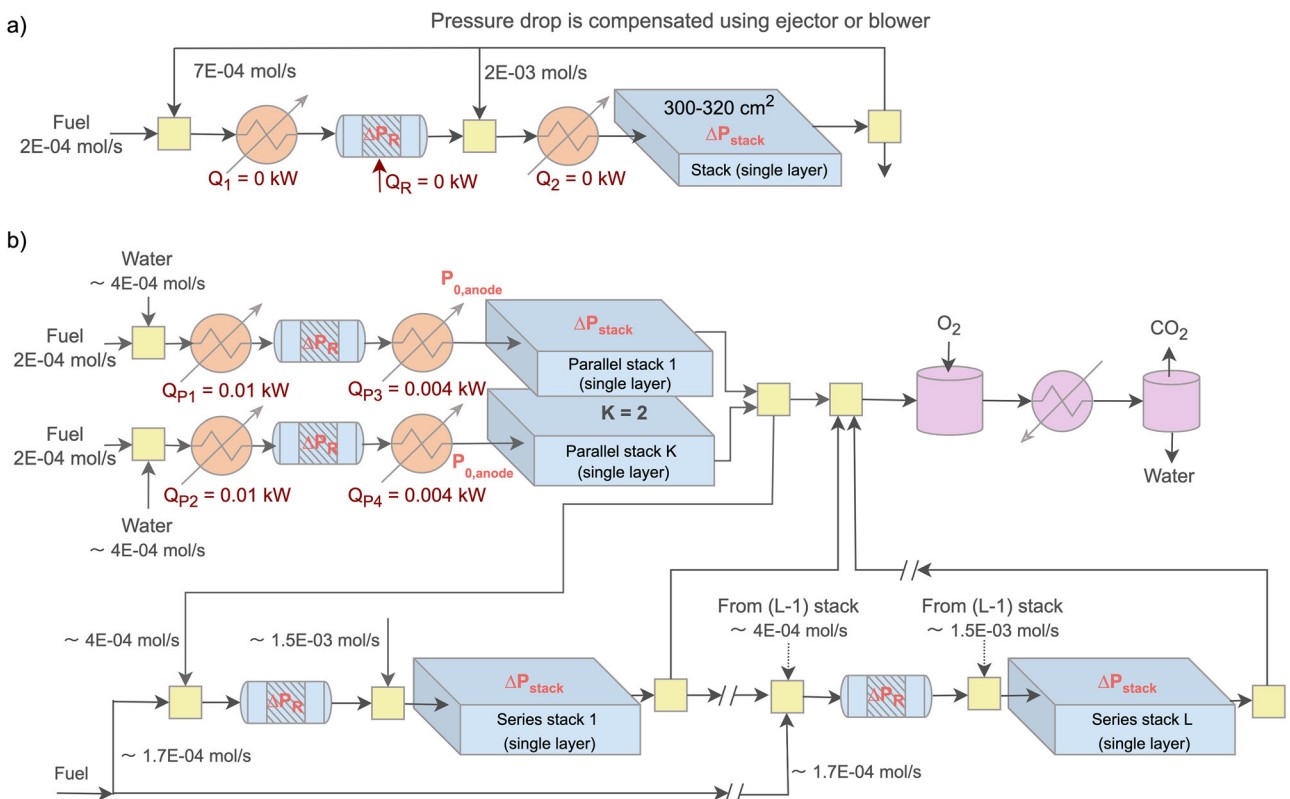

**Fig. 1 | Solid oxide fuel cell (SOFC) system configurations for enhanced anode off-gas utilization. a** Conventional design, and **b** hybrid design, based on 0.1 kW (300–320 cm²) single-layer stack (theoretical basis).

Instead of recirculating AOG back to its originating stack, it is directed from an upstream stack to a downstream one, assuming a pressure differential is maintained. This enables AOG from the preceding stack to mix with incoming fresh fuel for the subsequent reformer and stack, eliminating the need for an AOG blower to overcome pressure losses. Moreover, the steam content in the high-temperature AOG satisfies the steam-to-carbon (S/C) ratio, thereby removing the need for external water in the downstream reformer and stack. The pipeline configuration also eliminates the requirement for two heaters typically used to preheat the fuel-steam mixture, before the reformer and the stack. This hybrid design can be systematically replicated up to the $L$th stack, supporting improved scalability and efficiency. As shown in Fig. 1b, AOG flow from two single stacks operating in parallel is sufficient to supply multiple stacks in series. If needed, a portion of the AOG from the parallel stacks can be diverted to the burner. The distribution of AOG between parallel and series stacks can be optimized to meet system requirements and enhance global fuel utilization.

Overall, optimum AOG usage within the hybrid system design for scaling up involves two stacks operating in parallel, each with $X$ kW capacity (or a single stack with $2 \times X$ kW capacity). These are then connected to the subsequent stacks operating in series, each also with $X$ kW capacity. The rationale for adopting the configuration with two stacks operating in parallel and then connected in series is based on the thermal balance and the steam required by the subsequent reformer and the stack units. The total flow rate from the two parallel stacks ($2 \times X$ kW) represents the minimum value needed to effectively remove the heat exchangers positioned before the reformer and the stacks, ensuring adequate preheating and steam supply, improving system efficiency, and minimizing unnecessary thermal losses. Although configurations with more than two stacks in parallel are technically feasible, they would generate excess flow beyond what is required. This surplus flow would then need to be directed to the burner, where its chemical energy would be converted into heat.

In the proposed SOFC system design, the stack has to be decentralized, allowing for the use of standardized sizes produced by the manufacturer. Similarly, the reformer is also considered decentralized, with standardized sizes, as it is solely designed to provide syngas to the directly linked stack. Conversely, the burner is treated as a centralized component designed to collect and combust all unconverted fuel. However, this centralization does not imply that only one burner will be used in practical applications; the actual number of burners deployed depends on the availability of appropriate sizes in the market, which is discussed further in cost analysis.

The proposed hybrid design requires a detailed pressure analysis to determine the minimum inlet pressure ($P_0$) for the first two stacks operating in parallel. This pressure must be sufficient to overcome all downstream pressure drops across reformers, stacks, heat exchangers and burners while ensuring that the cooled exhaust gases remain slightly above atmospheric pressure for effective collection. Although elevated operating pressures can enhance SOFC system performance, they also introduce safety risks. Therefore, accurate estimation of the minimum required $P_0$ on both the anode and cathode sides is essential for safe and efficient system operation. Traditionally, pressure drops for various components in industrial settings are estimated based on empirical correlations. However, in this study, we employ a more fundamental approach, combined with literature-based data, to estimate pressure drops using a Monte Carlo framework. The intent is not to treat geometric or kinetic parameters as inherently random, but to conduct a broad uncertainty exploration across plausible operating and design ranges (e.g., flow rate, channel diameter, reactor height). Joint sampling over uniform ranges allows interactions among variables to be evaluated rather than only one-factor-at-a-time variation, helping to identify a realistic upper bound. We adopt the 95th percentile of the sampled pressure drop as a conservative design requirement to ensure adequate margin (see the subsection "Pressure drops estimation").

Three key components, namely reformer, stack, and burner system, exhibit crucial pressure drops that significantly influence the value of $P_0$. In stacks, pressure losses are determined by the fluid's viscosity, density, and flow distribution and can differ between the anode and cathode sides. It is important to note that each stack may comprise multiple layers, with the inlet flow distributed uniformly among these layers. As a result, the total pressure drop across a stack remains constant regardless of the number of internal layers. From the modeling results, the first two stacks, operating in parallel, exhibit an anode side pressure drop of 26.5 mbar. The subsequent stacks, operating in series, each experience a slightly higher pressure drop of 36.9 mbar due to the slightly higher flow rate and slightly different gas compositions, which does not bring a significant impact on the stack, as it is still far from the maximum stack pressure drop (100 mbar). Consequently, to simplify the analysis and ensure conservative estimations, a uniform pressure drop of 40 mbar ($\Delta P_S$) has been applied across all stacks.

The pressure drops across the reformer and burner system are primarily determined by catalytic reactor and combustor design principles. For the reformer, Monte Carlo simulations suggest that the pressure drop is most likely around 4 mbar, with a statistical maximum value of around 9 mbar. Therefore, to accommodate potential variations, a value of 10 mbar has been assigned for each reformer ($\Delta P_R$). The mean estimated pressure drop for the burner lies around 2 mbar, with the maximum value reaching 5 mbar. Due to the higher uncertainty in this estimation method, a conservative value of 15 mbar ($\Delta P_B$) has been used for the burner. Finally, for all burner downstream components, including the cooler and condenser, a pressure drop of 10 mbar is considered. More details on these pressure drop calculations can be found in the Supplementary Information (SI), Section B2 (Table B2.1 and Fig. B2.2).

Overall, Eq. (1) describes the pressure calculation in relation to the number of single-layer stacks connected in series, denoted by $L$. The inlet pressure at the anode side of the first two parallel stacks, $P_{0,anode}$, must be sufficient to overcome the cumulative pressure drops across all downstream components. To mitigate the risk of fuel leakage, $P_{0,anode}$ is typically maintained higher than $P_{0,cathode}$. Therefore, the cathode inlet pressure, $P_{0,cathode}$, is set based on the allowable pressure drop across the stack ($\Delta P_{max}$), as presented in Eq. (2).

$$P_{0,\text{anode}}(\text{bar}) = \Delta P_S + L \times (\Delta P_R + \Delta P_S) + \Delta P_B + P_{atm}$$
$$= 0.075 + 0.05 \times L + P_{atm} \tag{1}$$

$$P_{0,\text{cathode}}(\text{bar}) = P_{0,\text{anode}} - \Delta P_{max} \tag{2}$$

## SOFC hybrid design−COG recirculation heat cascade

Cathode off-gas (COG) recirculation is rarely explored in modern fuel cell system designs, as it involves complex challenges similar to those of AOG recirculation, including the need for specialized components such as high-temperature blowers or ejectors to overcome pressure drops. Despite these challenges, COG recirculation offers benefits such as improved heat management and reduced fresh air consumption. Although the energy consumption of a COG blower may account for approximately 5% of the power generated by the stack[28], optimizing COG recirculation could still enhance overall system efficiency. Thus, this subsection explores strategies for effective COG utilization, building on the findings from AOG configurations. For simplicity, the analysis is also presented using a single-layer stack; however, the system configuration is equally applicable to stacks with multiple layers, as both mass and energy balances scale linearly.

As two stacks operate in parallel and additional stacks are arranged in series (the AOG configuration from the previous subsection), three primary COG configurations can be identified, as shown in Fig. 2.

Initially, fresh air is compressed by a blower (B0); typically, the cathode air inlet pressure is maintained between 10 mbar and the maximum allowable pressure difference ($\Delta P_{max}$), below the anode fuel inlet pressure. This ensures safety by preventing leakage or explosion while maintaining a cathode outlet pressure above atmospheric pressure for proper exhaust release. After compression, the air is heated to the required stack inlet temperature in a heater (H0) before being distributed to the two parallel stacks, supplying the necessary oxygen and regulating stack temperature. In all three COG configurations shown in Fig. 2, the air flow rate and air processing module arrangement are consistent for the first two parallel stacks.

In the first simple series configuration (Fig. 2a), the COG flow is nearly halved at the cathode outlet of two parallel single-layer stacks; one part of this flow, containing diminished oxygen, is cooled in a cooler (C0) to the required stack inlet temperature and then directed to the first stack in series. This process is repeated between each pair of consecutive stacks in the series arrangement. The only constraint in this configuration is that the oxygen mole fraction in the depleted air (i.e., cathode outlet flow) from the $L$th stack in series must exceed a certain percentage, as specified by the stack manufacturer, to prevent any potential degradation. For example, if a minimum of 10% mole $O_2$ is required, then only three stacks can be linked in series ($L = 3$). Ultimately, depleted air from various stacks is collected and passed through the cooler CL. The system includes one heater (H0) and $L + 1$ coolers (C0, C1−CL). Additionally, since the cathode air inlet composition for each stack differs, the required flow rate may vary slightly, implying potential variations in pipelines. Overall, the maximum number of stacks in series is limited by the minimum allowable oxygen mole fraction at the cathode outlet specified by the manufacturer.

Figure 2b illustrates (the second) COG complex series configuration, where the airflow through the two parallel single-layer stacks remains identical to that in Fig. 2a. After supplying oxygen and balancing heat in the parallel stacks, a portion of the cathode outlet air is mixed with a controlled amount of fresh air. The fresh air is preheated to ensure that, after mixing, the resulting flow meets the temperature requirements of the next stack in the series. In principle, the cathode outlet air can be split in any ratio, and fresh air can be adjusted to meet flow and temperature demands, provided the oxygen mole fraction at the cathode outlet remains above a critical threshold to avoid degradation. Ideally, most of the cathode outlet air would be reused to minimize fresh air demand and reduce power consumption by the secondary blower (B1). This process could be repeated for each subsequent stack in series. The design imposes significant operational constraints, particularly due to the difficulty of managing fresh air flow. Any variation in fuel composition or stack performance requires real-time airflow adjustments, making the system highly sensitive to heat balance. This complexity complicates control and undermines the modularity essential for scalable SOFC design.

Figure 2c presents the third COG configuration, representing a hybrid parallel design. While the initial airflow to the two parallel single-layer stacks remains consistent with previous configurations, the depleted cathode outlet air is subsequently collected and mixed with fresh air at controlled temperature and pressure. This requires a dedicated arrangement involving a fresh air blower (B1) and heater (H1), which symbolically represent one or more units. The fresh air flow rate depends on the number of downstream stacks in series, with each requiring approximately 1.4E-02 mol/s. Heater (H1) ensures the mixed stream reaches the desired inlet temperature for even distribution. Although this configuration introduces additional air processing components, it eliminates the need for $M$ individual coolers compared to a simple series configuration, maintains a consistent cathode air inlet composition, and supports modular system design. Overall, the three COG configurations, two distributing cathode air in series and one in parallel, were evaluated based on manufacturing complexity, balance-of-plant (BoP) simplicity, and pipeline connection

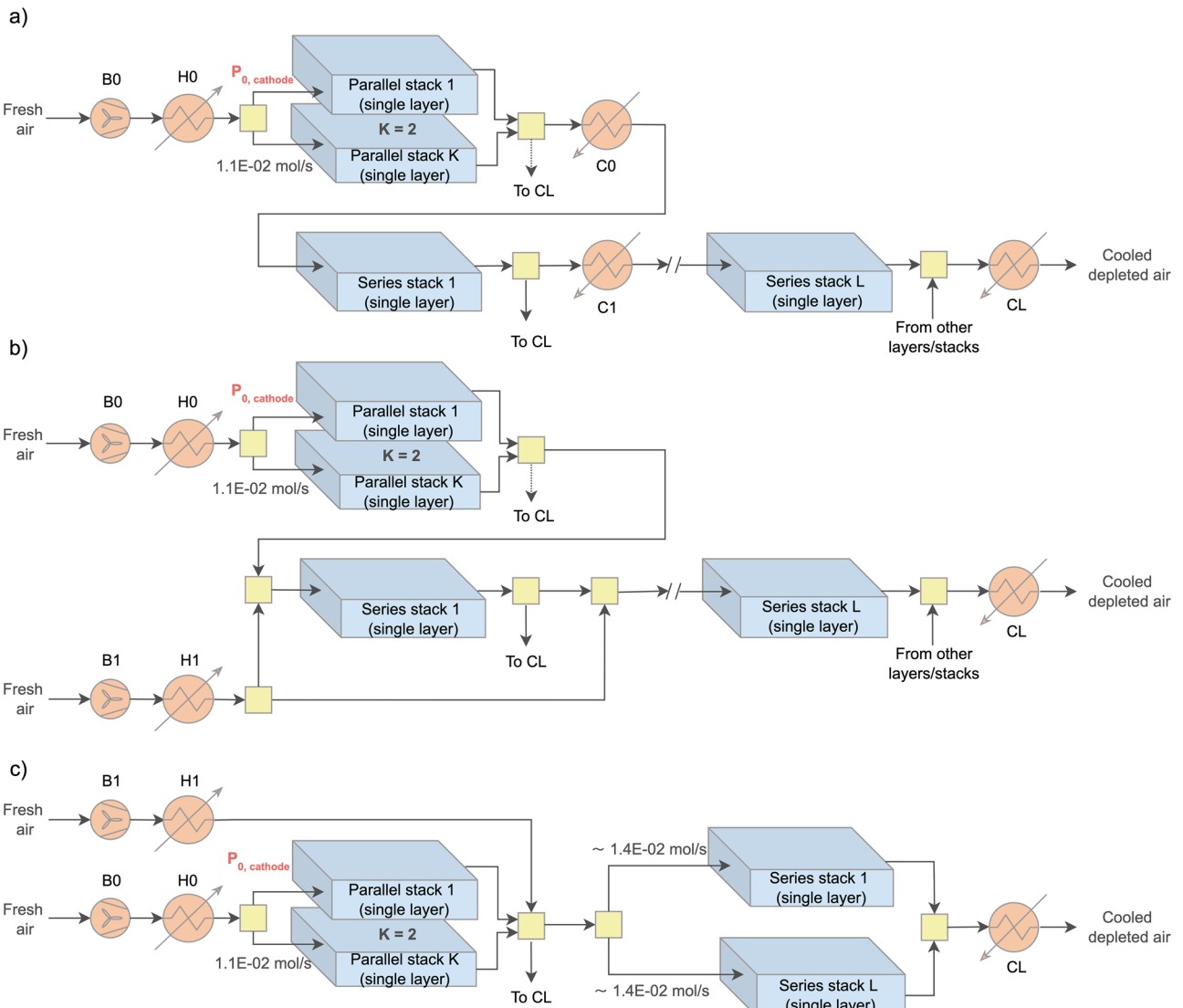

**Fig. 2 | Solid oxide fuel cell (SOFC) system configurations for enhanced cathode off-gas utilization. a** Simple series configuration, **b** complex series configuration, and **c** parallel configuration or hybrid design, based on 0.1 kW (300–320 cm$^2$) single-layer stack (theoretical basis).

requirements. Ultimately, the configuration shown in Fig. 2c was selected for the hybrid SOFC system scale-up due to its streamlined layout and advantages in modular integration.

In general, we examined various AOG and COG flow configurations without specifying parameters such as stack size, the number of layers per stack, or the overall system module scale. The objective was to develop a flexible and adaptable system layout that supports strategic decisions on the centralization or decentralization of component modules. This versatile framework allows industrial partners and academic researchers to tailor the system design to specific operational constraints, including maximum allowable pressure drops across stacks, minimum oxygen concentration at the cathode outlet, S/C ratio, and other relevant parameters.

### Case study on a 50 kW SOFC system

To illustrate the proposed hybrid design approach, the electrical performance of a 50 kW SOFC system is analyzed as a representative case. It is worth-mentioning that the values reported here correspond to a representative fixed operating point selected to enable a fair, like-for-like comparison between the proposed layout and the common designs. This should not be interpreted as a requirement to operate the plant at these exact setpoints. As indicated by the standardized

connection (plug-in) points shown as yellow boxes in Fig. 2b, the component modules can be reconfigured and rerouted to accommodate operating variations or emergency conditions without redesign. Moreover, although 50 kW remains relatively small compared to national electricity demand, this subsection focuses on comparing system performance. Therefore, the performance conclusions drawn here are scalable to larger systems. In subsequent subsections, where the emphasis shifts to economic analysis rather than efficiency, the system size will vary to match the specified global electricity output.

The analysis assumes a 10 kW stack capacity, consistent with current market availability, a maximum allowable pressure drop of 100 mbar across the stack[14,29–31], and a minimum oxygen concentration of 15% at the cathode outlet[32–34]. The layout and number ratio of standardized modules for the proposed hybrid SOFC design are depicted in Fig. 3a. The number ratios in parentheses (e.g., ×1, ×2, ×3) indicate the relative number of each module required in one complete system. For example, one fuel/water module supports two parallel stack modules and three series stack modules.

This ratio can be proportionally scaled for larger system capacities. For the 50 kW system configuration, it involves two 10 kW stacks operating in parallel ($K = 2$) and three in series ($L = 3$), which results in a pressure requirement of less than 1.23 bar across the stack modules.

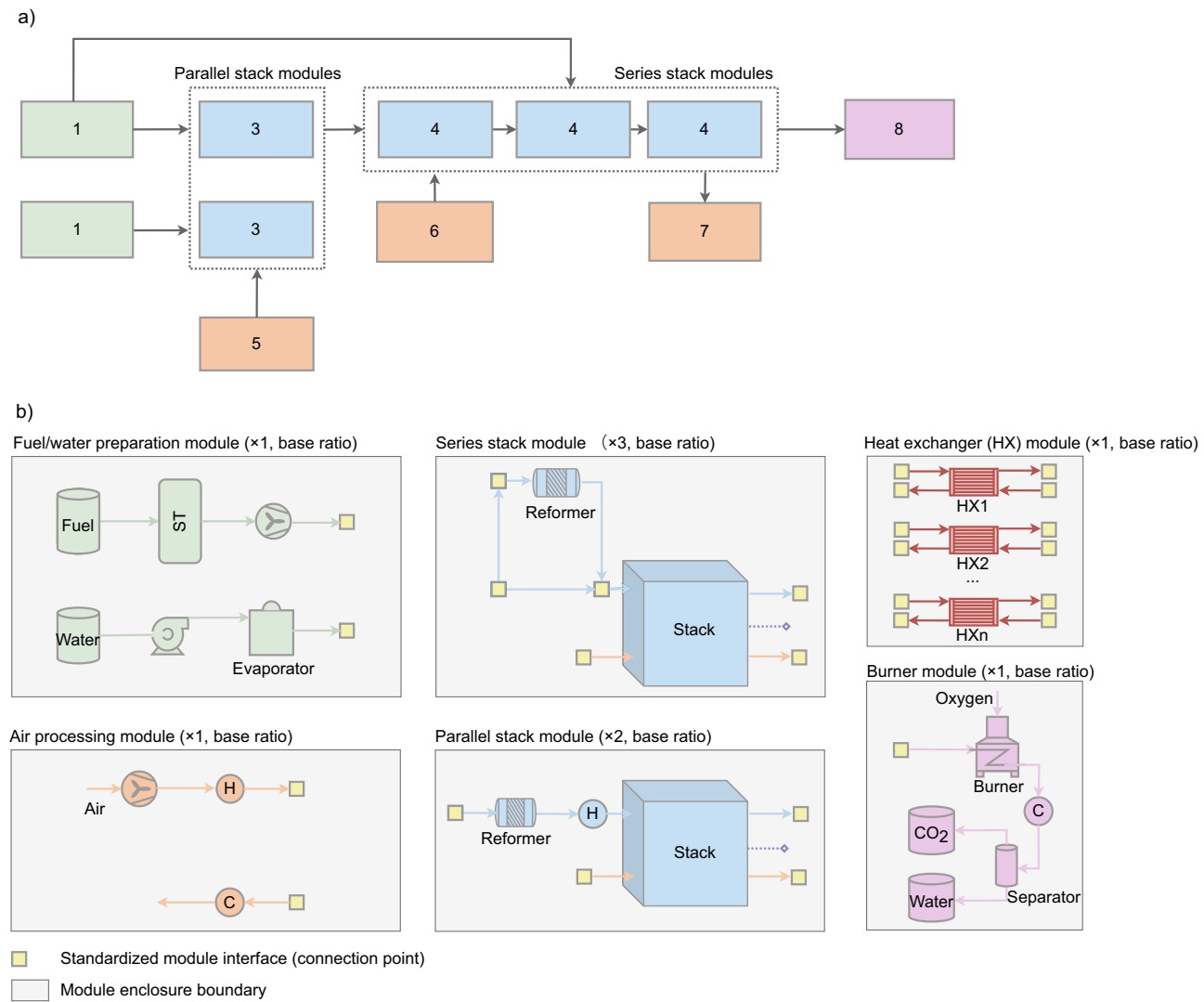

**Fig. 3 | Modular layout of a 50 kW solid oxide fuel cell (SOFC) system. a** Layout of a 50 kW SOFC system based on the hybrid design approach, and **b** layout of various modules.

This value is within the design tolerance of several commercial SOFC stacks, including those from SolydEra[14,35,36], which can safely operate at modest pressure differences when properly sealed and supported. It is important to note, however, that when more than three stacks are connected in series, the cumulative pressure drop increases and requires verification through manufacturer-specific testing or design adjustment. While fully serial multi-stack configurations are not yet common in commercial systems, the partial concept has been demonstrated at pilot scale[37–40] and remains technically feasible for modular systems with controlled pressure balancing. The present work, therefore, proposes a conservative upper limit for the number of series-connected stacks that ensures operational safety while maintaining compact system integration.

This setup requires the production of six standardized modules, shown in Fig. 3b. Centralized modules in the proposed hybrid design include (i) fuel and water preparation module; (ii) an air processing module, which compresses and heats fresh air for the cathodes and cools the collected air from multiple stacks; (iii) a heat exchanger module managing thermal exchange between hot and cold flows via variably sized exchangers; and (iv) a burner and downstream BoP module that collects and combusts unconverted fuel, followed by $CO_2$ and water separation after cooling. In addition, two types of decentralized stack modules are defined, with their module number

varying according to the size of the SOFC system. The first, used in the parallel arrangement, comprises a stack, an external reformer, and a heater. It requires both fresh fuel and an external water supply due to the absence of AOG recirculation. The second, used in the series arrangement, includes a stack and an external reformer, which utilizes both fresh fuel and the AOG flow from the preceding stack module.

In the proposed layout (Fig. 3a), each stack module can either be enclosed in a thermally insulated hotbox[41–43] or integrated within a shared enclosure/hotbox[38,44,45], depending on design preference and system scale. The independent-hotbox choice offers higher modularity and operational flexibility, as it allows local thermal control for each stack, minimizes thermal cross-interference, shortens intra-module piping (reducing heat loss and pressure drop), and simplifies maintenance or $N+1$ redundancy through plug-in replacement. Conversely, integrating multiple stack modules in a single large hotbox can simplify insulation, reduce external heat losses, and lower the total number of mechanical interfaces; however, it also makes it difficult to cool individual stacks while others remain hot, due to elevated internal air and radiative temperatures. Furthermore, such shared enclosures often increase overall system volume to maintain adequate thermal spacing. In this study, we focus on the modular or independent-hotbox approach as a baseline concept[46–50], but both approaches remain

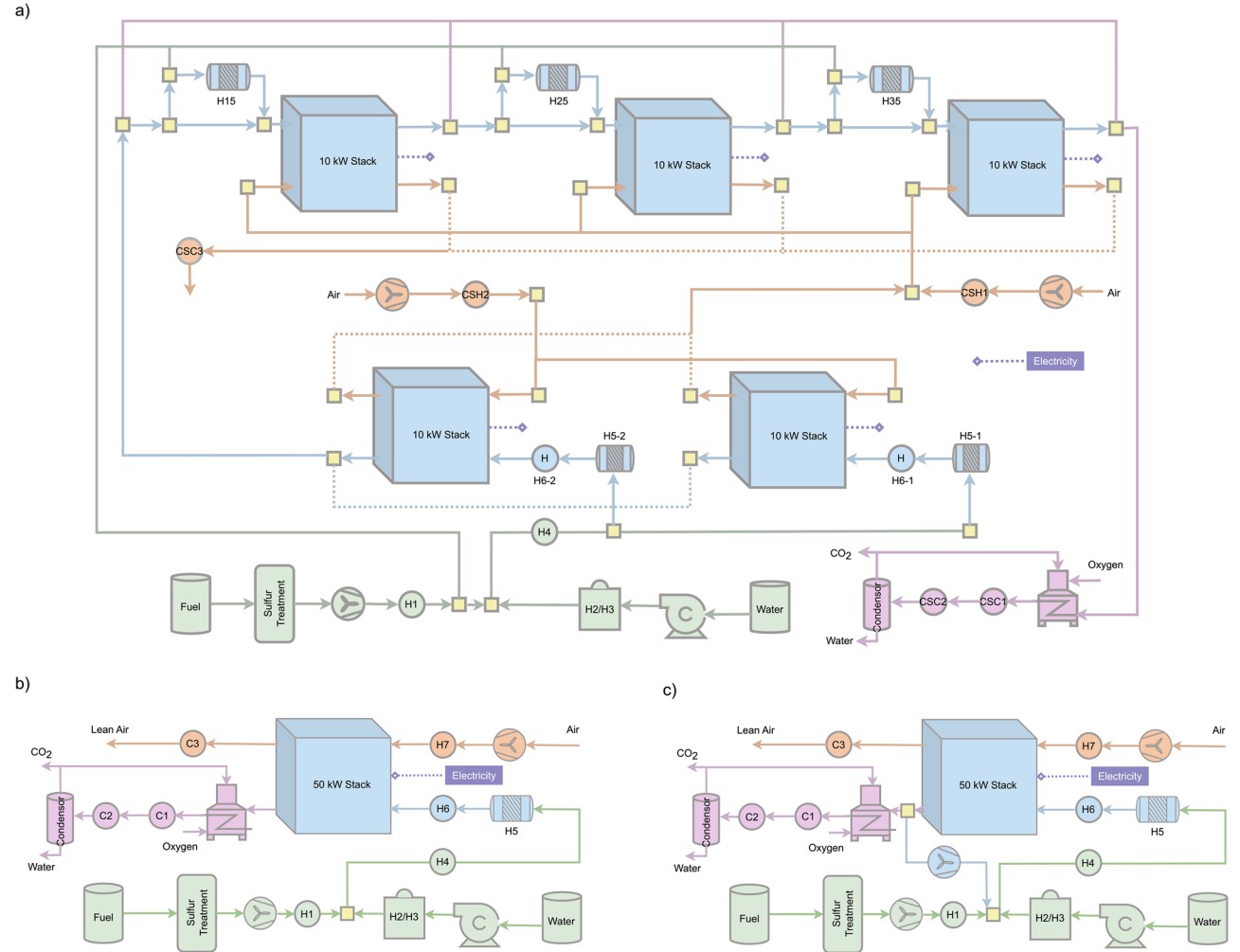

**Fig. 4 | Solid oxide fuel cell (SOFC) configurations delivering 50 kW net electrical output. a** Proposed hybrid design (FCFA): five 10 kW stacks arranged - two in parallel followed by three in series; **b** centralized design without recirculation (NCNA): illustrated with five 10 kW stacks in parallel; an equivalent realization using a single 50 kW stack is also feasible; **c** centralized design with hot anode off-gas (AOG) recirculation (NCHA): likewise realizable with five 10 kW parallel stacks or a single 50 kW stack.

technically feasible and may be adopted depending on the manufacturer's strategy and plant size.

In the event that a module temporarily fails or its upstream counterpart stops providing sufficient flow, the system design preserves self-sufficiency through built-in provisions. Each stack module design has fixed interfaces for fresh fuel, steam, and process air; the yellow boxes in Fig. 3b indicate standardized connection (plug-in) points that allow any module to be coupled to any other. In such contingencies, the missing input is primarily thermal. This shortfall can be covered either by activating a small electrical heater which maintains the reformer and stack at required temperatures during brief interventions and is supported by $N + 1$ redundancy for rapid hot-swap, or by routing a dedicated start-up heat exchanger in the centralized heat exchanger module to deliver supplemental duty during transients or emergencies. These provisions maintain operation without a full plant shutdown and prevent an energy-deficient state from propagating downstream. Detailed transient control and digital-twin supervision lie beyond the scope of this steady-state study, but it is an interesting and important area that can be addressed in future work. Finally, it is important to clarify that, although the proposed scale-up strategy fixes the interconnection geometry, it increases system-level flexibility by enabling capacity right-sizing from standardized modules

and by simplifying maintenance and replacement, thereby offering potential practical insights for industrial manufacturing partners.

In Fig. 4, the heaters/coolers indicate required duties rather than specific components. In practice, these services can be provided by standardized heat-exchanger modules; multiple (hot and cold) process streams are routed to a centralized module that contains the necessary heat-exchangers sized per duty. This approach preserves the plug-and-play philosophy (fixed interfaces, common headers) while allowing the heat-exchanger module to be the primary element that is tailored to plant scale. The concept is consistent with ongoing industrial efforts toward standardized and modular SOFC packages[32,51–55].

Figure 4a illustrates the detailed process flow diagram of a 50 kW SOFC system employing the proposed hybrid design strategy (i.e., forward COG and forward AOG, FCFA). While the component modules are standardized, their operating conditions may differ, particularly for stacks configured in series. Therefore, system-level optimization has been performed to enhance overall performance. Although achieving and maintaining optimal operating conditions in large-scale systems can be challenging due to operational fluctuations, assessing the energy efficiency potential of the proposed design provides valuable insights into system performance prior to economic evaluation. Accordingly, this subsection presents multi-objective optimization

**Table 1 | Objective functions, decision variables, and selected flows for the hybrid design (50 kW SOFC system); performance comparison among three designs for a 50 kW SOFC system**

| Objective functions | | | | |
|---|---|---|---|---|
| Electrical efficiency, % | 66.3 | | | |
| Global fuel utilization, - | 0.98 | | | |
| External water flow rate, mol/s | 0.085 | | | |
| **Parameters** | **Parallel stacks** | **Series stack 1** | **Series stack 2** | **Series stack 3** |
| Reformer temperature for pre-reformer, °C | 539 | 533 | 518 | 519 |
| Reforming ratio for pre-reformer | 0.12 | 0.13 | 0.12 | 0.11 |
| Natural gas flow rate, kg/s | 0.0007 | 0.0003 | 0.0003 | 0.0003 |
| Natural gas flow rate, mol/s | 0.0409 | 0.0174 | 0.0187 | 0.0186 |
| External water flow rate, mol/s | 0.085 | 0 | 0 | 0 |
| Fresh air input, mol/s | 2.151 | 0.778 | 0.644 | 0.62 |
| COG air input, mol/s | 0 | 0.694 | 0.694 | 0.694 |
| Air output $O_2$ mole fraction, - | 0.184 | 0.178 | 0.175 | 0.174 |
| Air blower power, kW | 0.74 | 0.64 | | |
| Net power from module(s) | 20.88 | 9.89 | 9.95 | 9.79 |
| | **Hybrid** | **NCNA** | **NCHA** | |
| Natural gas flow rate, kW | 76.67 | 81.98 | 79.77 | |
| External water flow rate, mol/s | 0.085 | 0.212 | 0.085 | |
| Fresh air flow rate, mol/s | 4.193 | 5.377 | 6.42 | |
| Electricity output, kW | 50.51 | 52.79 | 45.9 | |
| Electrical efficiency, % | 66.3 | 65.8 | 57.5 | |
| Heat available at 600 °C, kW | 4.1 | 7.93 | 5.66 | |
| Heat available at 200 °C, kW | 4.47 | 0.92 | 8.87 | |
| Cold utility requirement, kW | 30.77 | 37.4 | 40.25 | |
| Additional electricity generated by RC, kW | 2.03 | 2.14 | 2 | |
| Electrical efficiency with RC, % | 68.5 | 67 | 60 | |
| Direct mixing points at high temperature | 8 | 0 | 5 | |

COG cathode off-gas, RC Rankine cycle.

analysis results, as described in the subsection "Operating conditions optimization," to explore the achievable electrical efficiency of the hybrid design. Furthermore, the hybrid system design is compared with two conventional fully centralized configurations, NCNA (no COG and AOG recirculation) and NCHA (no COG and hot AOG recirculation), as shown in Fig. 4b, c, to evaluate the advantages and limitations of the proposed hybrid design approach.

Table 1 presents the primary material and energy flows for the proposed hybrid system design, highlighting the effectiveness of AOG utilization. The fresh fuel requirement for the two parallel stacks (20 kW total) is 0.0007 kg/s, with each stack requiring 0.00035 kg/s. In contrast, the stacks operating in series (10 kW each) require approximately 0.0003 kg/s, which underscores the benefit of AOG utilization. Additionally, AOG recirculation significantly reduces the external water requirement, with a total of 0.085 mol/s needed for two parallel stacks and none for the series stacks. Similarly, COG recirculation contributes to a reduction in fresh air requirements (less than 0.778 mol/s for parallel stacks compared to 1.076 mol/s for series

stacks) while maintaining the minimum $O_2$ mole fraction at the cathode outlet. Furthermore, the optimization results, as shown in Table 1, demonstrate that decentralized reformers can operate at varied temperatures and reforming fractions, further validating the modularity concept.

The hybrid design requires some stacks to operate at slightly elevated pressures to enable efficient utilization of AOG and COG. In general, the anode side benefits from the higher $CH_4$ storage and supply pressure. A notable drawback is the increased power consumption of air blowers serving both parallel and series stacks. This raises a critical consideration regarding whether the advantages of AOG and COG utilization are sufficient to offset the additional energy demand for air compression. As shown in Table 1, the hybrid design reduces air flow requirements by 22% and freshwater consumption by 59.9% compared to the conventional NCNA SOFC configuration. Assuming a maximum pressure drop of 100 mbar across the stack, in line with recommendations from EU projects[32,56], the hybrid system design achieves an electrical efficiency of 66.3%, surpassing the 65.8% of the NCNA design. Although the NCHA design exhibits similar water usage to the hybrid design, its electrical efficiency is significantly lower at 57.5%, primarily due to reliance on a low-efficiency, high-temperature AOG blower.

In addition to electrical efficiency, SOFC systems are valued for their heat recovery potential. Due to its higher electrical efficiency and optimized high-temperature operation, the hybrid design co-generates less high-temperature heat (4.10 kW) compared to the NCNA (7.93 kW) and NCHA (5.66 kW) designs. Nevertheless, it still produces 4.47 kW of medium-temperature heat (at around 200 °C), which is suitable for residential heating or industrial applications such as dairy processing. An alternative use for this heat is integration into a Rankine cycle (RC)[28], which has also been evaluated. With RC integration, the hybrid system's electrical efficiency increases to 68.5%, compared to 67% for NCNA and 60% for NCHA. Detailed heat flow data for all three system designs are provided in the SI, Section B1, Table B1.1.

After the system performance analysis, it is important to explore trade-offs among scale-up strategies. For the proposed hybrid strategy, the stand-alone electrical-efficiency gain (without a Rankine cycle) is modest, about 0.5% relative to the NCNA layout, which understandably raises the question of whether off-gas routing is worthwhile. However, three additional benefits can be listed as follows. (i) Water and air demand: The hybrid layout reduces external water make-up by 59.9% and fresh-air flow by 22%. In most regions, make-up water must be purified to stack-grade quality, adding non-negligible CAPEX and heat requirement[57–61]; lowering the make-up rate therefore reduces both cost and auxiliary loads. (ii) Heat-grade matching: By shifting heat recovery from high to medium temperature, the hybrid design provides less heat at 600 °C (4.10 vs. 7.93 kW) but more at ~200 °C (4.47 vs. 0.92 kW), which better matches typical industrial and district-heating requirements; this avoids the thermodynamic penalty of cooling high-grade heat just to use it at low temperature. (iii) Rankine cycle integration: With a simple Rankine cycle, the net electrical efficiency improvement rises to 1.5% over NCNA (68.5 vs. 67.0%), amplifying the value of medium-temperature heat available. The latter subsection shows that these resource savings and heat-utilization advantages translate into a lower levelized cost of electricity (LCOE) at the large scale, even when small-scale efficiencies are similar. Finally, we note that most current commercial SOFC deployments resemble the NCNA configuration because of its simplicity and control robustness; our contribution is to show how a modular series-parallel architecture can capture the above system-level benefits while remaining manufacturable through standardized sub-modules.

Overall, through this subsection, we have demonstrated that the proposed hybrid design improves both electrical and thermal efficiency. However, it is crucial to emphasize that efficiency should not be

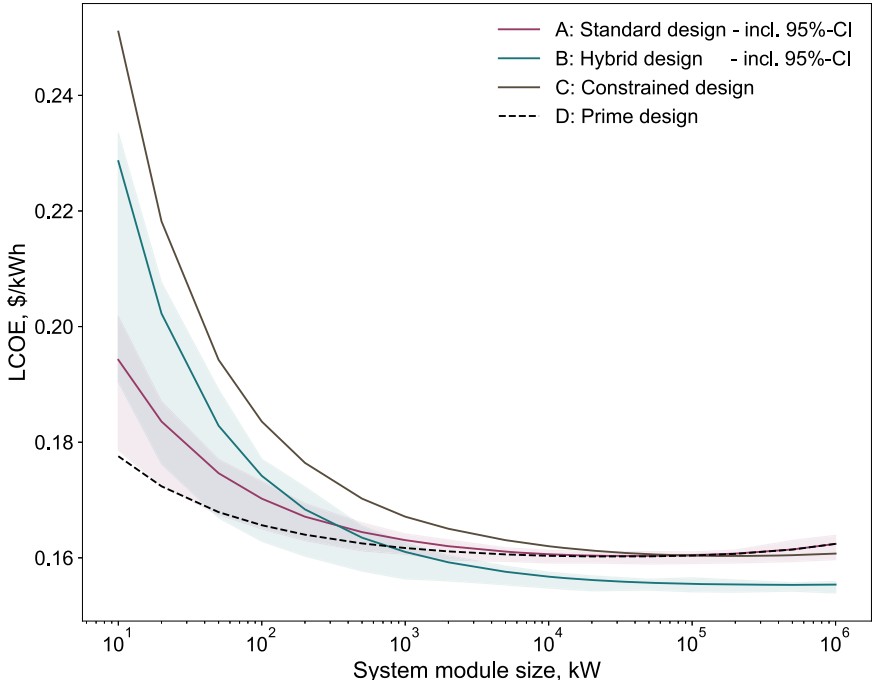

**Fig. 5 | Levelized cost of electricity for modular solid oxide fuel cell (SOFC) scale-up.** Levelized cost of electricity comparison across scale-up strategies and system module sizes (1 GW yearly power production) for a nominal fuel price of 10 cents/kWh. Source data are provided in "Source Data.xlsx".

the sole criterion in system design. While the 50-kW system was selected to illustrate the performance advantages of the proposed hybrid design over conventional designs, this scale is modest relative to national electricity capacities. Therefore, the following subsection presents a detailed techno-economic analysis aimed at assessing the cost benefits of the proposed scale-up strategy without constraining the system to a fixed module capacity. The primary objective is to evaluate whether this modular approach can enable the cost-effective scale-up of energy storage systems on a global level.

**Techno-economic analysis background**

In this subsection, we have presented a comparative techno-economic analysis of the four scale-up strategies for the SOFC system: standard system design, hybrid system design, constrained system design, and prime system design, and the differences among all strategies are explained in the subsection "Overview of scale-up strategies." The analysis is conducted under a fixed global electricity output of 1 GW, while the system module size varies from 10 kW to 1 GW. It is important to clarify that a single 1 GW SOFC module is not intended as a realistic present-day design point. We include the full continuum of module sizes up to 1 GW purely as an upper-bound envelope to reveal asymptotic cost trends (economies of scale vs. production volume) and to avoid biasing conclusions by today's market availability. In practice, some components cannot yet be built at very large single-unit sizes, and the practical limit varies by supplier and price. Those limits are partly technical (true scale-up challenges) and partly manufacturing market constraints that can evolve with standardization and volume[62–66]. Our goal here is to show the direction and magnitude of cost reduction as module size increases, not to prescribe a 1 GW single module.

Uncertainty analysis is also included to assess the confidence in the obtained results. The evaluation begins with the crucial metric, LCOE, to observe its variation across different strategies and scales. This is followed by a detailed breakdown of capital expenditure (CAPEX) and a sensitivity analysis of the fuel price, the main contributor to operating expenditure (OPEX), to explain the underlying

drivers for the observed LCOE trends. Overall, this subsection aims to highlight the trade-offs, advantages, and limitations associated with each scale-up strategy.

**Levelized cost of electricity trends across scale-up strategies**

Figure 5 presents the LCOE trends as a function of system module size for the four scale-up strategies. In all cases, LCOE shows a sharp initial decrease with increasing system module size, reflecting classical economies of scale. This trend saturates at large scales, where other factors come into play. For the standard system design strategy A, LCOE declines from 0.194 $/kWh for the 10 kW system module to 0.162 $/kWh for the 1 GW system module. A similar trend is observed for the prime system design strategy D, with LCOE ranging between 0.178 and 0.162 $/kWh. The lower LCOE for strategy D at small scales is attributed to its flexibility in deploying larger centralized BoP components ($P_{rest}^D$ in Eq. 15). As system module size increases beyond 10 MW, strategies A and D converge, both approaching an LCOE of 0.162 $/kWh. This convergence occurs because, at large scales, the distinction between centralized and modular designs diminishes.

For the hybrid system design strategy B, the LCOE also decreases with increasing system module size, ranging from 0.229 $/kWh for the 10 kW system module to 0.155 $/kWh for the 1 GW system module. At smaller scales, strategy B yields higher LCOE values than strategies A and D due to the inclusion of multiple small-size, low-technology readiness level (TRL) component modules (i.e., stacks and reformers) within each system module. Since large-size component modules are generally more cost-effective than several small-size component modules of equivalent capacity, this consideration increases LCOE at small scales for strategy B. However, the LCOE for strategy B declines more rapidly with scale, outperforming the LCOE of strategy A beyond a system module size of approximately 300 kW, as strategy B has higher electrical efficiency (see Table 1). This finding suggests that the standard design strategy may be more economical for SOFC systems below 300 kW, while the hybrid design strategy offers superior performance for larger-scale deployments, ultimately achieving the lowest LCOE of 0.155 $/kWh for the 1 GW system module, which is also compatible with expectations[67–69].

Finally, the constrained system design strategy C, which builds upon the concept of strategy B, includes a greater number of small-sized component modules not only for low-TRL components but also for the high-TRL BoP components within each system module. This high degree of decentralization results in the highest LCOE across all strategies, particularly at smaller system module sizes. However, as the system module size increases, the difference in LCOE between strategy C and the other strategies diminishes. At large scales, the LCOE for strategy C nearly converges with that of strategies A and D, reaching a minimum value of 0.160 $/kWh at the 1 GW system module.

Figure 5 indicates that the hybrid system design strategy B performs better at large scales, while the standard system design strategy A and the prime system design strategy D are more economical at smaller scales. However, the figure also raises several interesting questions worth examining in more detail. For example, strategies A and D generally show decreasing trends in LCOE and reach a stable value for larger module sizes, and there is an increase for the 1 GW system module. In addition, the shaded uncertainty bands reveal that the LCOE has a wide variation at small scales but becomes much narrower as the system module size increases. Finally, it is also important to understand why the LCOE difference among all strategies decreases as the system module size grows and why strategy B eventually becomes the most cost-effective despite starting out as the second most expensive at 10 kW. All of these questions are addressed in the following subsections

## Cost structure and sensitivity analysis

Figure 6a presents the normalized CAPEX contributions from various component modules across the four scale-up strategies for varying system module sizes based on a fixed annual electricity output of 1 GW. As expected, for small-scale system modules (e.g., 10 kW), the hybrid system design strategy B exhibits the second-highest CAPEX, driven primarily by the decentralized stack and reformer modules (1772 and 1731 $/kW, respectively), with one-fifth of the system module capacity. In contrast, the standard design strategy A features significantly lower costs for these component modules, with stack and reformer modules priced at 916 and 221 $/kW, respectively. For the constrained system design strategy C, the main cost difference relative to strategy B arises from the high-TRL BoP components, such as heat exchangers, due to the fivefold increase in their numbers per system module. This highlights the importance of maintaining some degree of centralization for specific BoP components, even within modular designs. The prime system design strategy D achieves the lowest CAPEX module cost, totaling 1359 $/kW, using the 10 kW case as an example, mainly due to all BoP components being fully centralized and sized for the entire 1 GW capacity.

As the system module size increases, the CAPEX gap among all strategies narrows significantly. By the time the system module size reaches 1 MW, the CAPEX difference among various strategies falls below 350 $/kW. Since stack and reformer modules remain the dominant CAPEX contributors in strategies B and C, this suggests substantial room for future cost reductions through technological advancements in these low-TRL components, an effect that would have a lesser impact on strategies A and D. Interestingly, for both strategies A and D, when the system module size reaches 1 GW (i.e., only one system module is needed to meet the global electricity output), the CAPEX contribution from reformer module increases compared to smaller system module sizes (e.g., 1 MW). This observation motivates a deeper investigation into the CAPEX behavior of low-TRL components for different sizes.

Unlike mature component modules such as heat exchangers, burners, and blowers, which are widely used across multiple industries, stack and reformer modules are specific to fuel cell systems and are less commercially mature. Their CAPEX is influenced by two competing factors: economies of scale and production volume. While a greater

component size intuitively implies lower normalized costs, it also directly affects the annual production volume for a fixed plant capacity. These opposing trends suggest the existence of a trade-off point, beyond which further scaling up of individual units no longer yields cost savings.

To investigate this, a sensitivity analysis was performed on global electricity output levels and the sizes of component modules, specifically stacks and reformers. The analysis considered three electricity output targets: 10 MW, 100 MW, and 1 GW, each under varying component module sizes. As shown in Fig. 6b, the CAPEX for reformer initially decreases with increasing module size but begins to rise around 1 MW. For instance, under the 10 MW plant output scenario (purple line), larger module sizes lead to lower production volumes, which in turn increases the unit cost. It is important to clarify that component module sizes above 10 MW are not displayed for this scenario, as the number of modules would be smaller than 1. Similar trends are observed for the 100 MW (green line) and 1 GW (orange line) plant output scenarios. Another key observation arises when the component module size is fixed at 10 MW and CAPEX trends are compared across all three global electricity output cases. CAPEX is highest for the 10 MW case and progressively lower for 100 MW and 1 GW systems. This is due to higher production volumes resulting from larger total plant output.

In contrast, the CAPEX of the stack (Fig. 6c) decreases consistently with increasing module size across all production targets (10 MW, 100 MW, and 1 GW), indicating that economies of scale prevail regardless of production volume. This highlights a key difference in the cost behavior of these two low-TRL technologies. Stack cost reduction is primarily a function of unit size, while reformer cost must be optimized by balancing unit size and annual production volume. This insight provides guidance for future research and development. Stack innovation should prioritize scaling-related challenges, such as thermal management at larger cell sizes, while reformer design should aim to optimize both scale and manufacturability to ensure cost competitiveness across deployment scales.

Figure 5 not only presents the LCOE trends across four scale-up strategies but also includes shaded regions around strategies A and B, indicating the results of uncertainty analysis on key input parameters. The uncertainty overlap for strategies A and B, particularly at smaller system module sizes, highlights the need to investigate the primary contributing factors. To better understand these drivers, a sensitivity analysis on LCOE was conducted for the hybrid design strategy as a case study. As shown in Fig. 6d, each key input parameter was varied individually, including the CAPEX of the stack, reformer, and BoP, as well as their associated lifetimes, the discount rate, and the fuel price. Among these factors, fuel price exhibited the most pronounced effect on LCOE, shifting the value between 0.14 and 0.22 $/kWh within the defined uncertainty range.

To further investigate the influence of fuel price on LCOE, a broader fuel price range from 0 to 25 cents/kWh was examined, divided into six discrete scenarios, as illustrated in Fig. 6e. For each fuel price, a separate LCOE curve is presented. In addition, for each curve, the uncertainty associated with other key input parameters is still considered through a multi-parametric sensitivity analysis. These uncertainties are represented by shaded areas surrounding each LCOE trend line. This approach captures a realistic range of future market conditions, including initial fuel price increases due to transitions from fossil fuels to renewable alternatives (e.g., natural gas to biomethane), followed by potential long-term cost reductions as supply stabilizes.

The results from Fig. 6e indicate that when fuel is assumed to be free (0 cents/kWh), the LCOE is uniquely driven by CAPEX, with capital costs being most prominent at smaller system module sizes and becoming less significant at larger scales, consistent with the trends observed in Figs. 5 and 6a. However, as fuel prices increase (the main driver in OPEX), the LCOE rises significantly. Across the system module

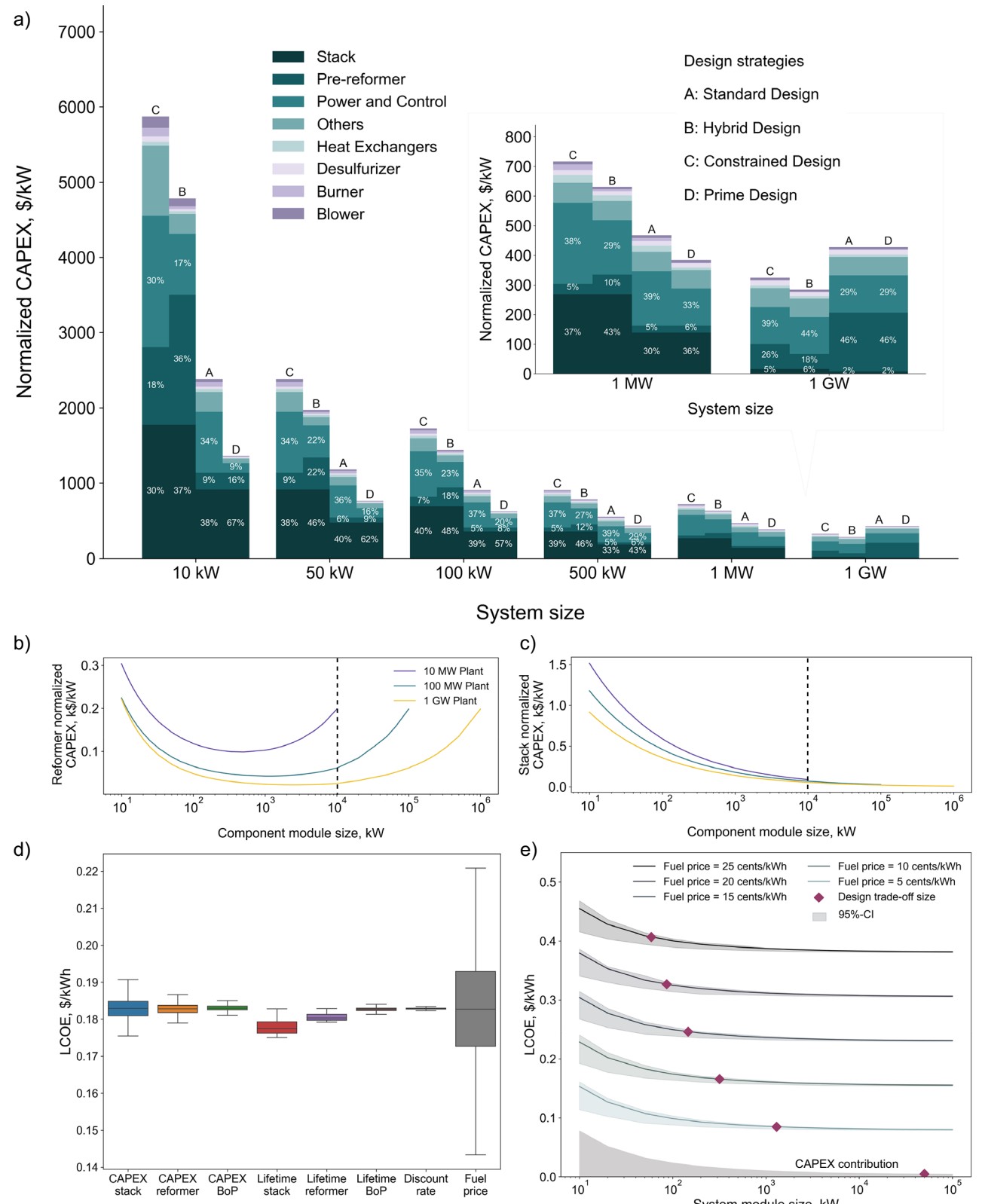

**Fig. 6 | Techno-economic analysis of modular scale-up strategies. a** Normalized capital expenditure (CAPEX) contribution of component modules across scale-up strategies. **b, c** Reformer against stack's CAPEX trends under varying annual production targets and module sizes. **d** Uncertainty analysis of eight input parameters and **e** sensitivity analysis on the fuel price with trade-off sizes for the hybrid design strategy. Source data are provided in "Source Data.xlsx".

sizes, the share of OPEX within the LCOE increases from approximately 50 to 95%, with an increase in the fuel price from 5 to 25 cents/kWh. Notably, at higher fuel prices, the trade-off LCOE point between the hybrid system design strategy B and the standard system design

strategy A shifts toward smaller system module sizes. This shift reflects the superior electrical efficiency of the hybrid design, which becomes economically advantageous even at smaller system module sizes under high fuel prices. These trends again explain the steeper LCOE

decline observed for strategy B in Fig. 5 and why it ultimately achieves a lower minimum LCOE of 0.155 $/kWh compared to the 0.16 $/kWh for strategy A. Finally, a sensitivity analysis was conducted on possible size limitations of low-TRL components by considering the current market maximum component size, which has shown its limited impact on the overall CAPEX, see Fig. B3.1 in SI. This finding aligns with the broader conclusion that efficiency becomes the dominant factor during system scale-up, underscoring a key direction for future development in both academic research and industrial applications.

### Summary of techno-economic analysis

The economic analysis reveals that the design strategy and system module size strongly influence LCOE. At small system scales, decentralized strategies, particularly hybrid and constrained system designs, exhibit higher LCOE due to increased CAPEX from multiple small-sized component modules. As system module size increases, the CAPEX gap among strategies narrows, and the system efficiency becomes dominant, making hybrid system design strategy B the most cost-effective design at system scales above 300 kW, ultimately reaching the lowest LCOE among the four system design strategies. Sensitivity analyses confirm that fuel price is the dominant factor, with OPEX contributing up to 95% in LCOE at high fuel prices. These findings underscore the importance of balancing component module size, production volume, and fuel conversion efficiency in optimizing SOFC scale-up strategies.

### Rationale and implications of the modular scale-up strategy

This study shows that the proposed hybrid design strategy performs well from both a system-analysis and an economic perspective. It also specifies a fixed module layout, piping interfaces, and module types. This reduces degrees of freedom and increases inter-dependence among stacks, but it addresses a major practical barrier to SOFC deployment. Today, many projects are engineered case by case (custom reformer and heat-exchanger sizing, redesigned piping, repeated P&ID reviews), which slows commissioning and raises cost. Standardization is therefore not a limitation but the core mechanism to accelerate adoption at scales relevant for long-duration and seasonal storage.

Concretely, standardized module production can (i) shorten lead times and lower engineering hours by shifting effort from bespoke plant design to repeatable module fabrication, (ii) enable learning-curve cost reductions through higher production volumes, (iii) provide system-level flexibility by allowing capacity to be built from interchangeable modules rather than one-off plants, and (iv) improve maintainability and availability. Failed modules can be isolated and swapped with spares, reducing downtime and service cost.

We acknowledge open questions around control, maintenance, and fault management. These can be mitigated by standard control interfaces and setpoints across modules, isolation valves, and bypass manifolds to decouple modules during transients or faults, conservative pressure-drop budgets to preserve operability, and the option to hold one module in hot standby ($N+1$ redundancy) to fulfill the availability targets. Our aim is not to claim that a single layout is universally optimal, but to provide a reproducible, manufacturable reference point that component suppliers can jointly adopt. Such coordination is a critical step if SOFC-based long-term storage is to scale globally in support of net-zero objectives.

## Discussion

In this study, we presented a modular hybrid design strategy for scaling up SOFC systems, emphasizing the strategic integration of AOG and COG recirculation. The proposed architecture balances the centralization and decentralization of key component modules through a flexible series-parallel configuration, enabling improved system efficiency, reduced reliance on external resources, and simplified integration.

A detailed case study demonstrated that the hybrid system can achieve an electrical efficiency of 66.3%, reduce external water use by 60%, and lower fresh air demand by 22% compared to a baseline centralized design (NCNA). When integrated with a Rankine cycle, the total electrical efficiency increased to 68.5%, exceeding both NCNA (67%) and NCHA (60%) configurations. These improvements are accompanied by medium-temperature heat recovery potential, further enhancing the system's cogeneration capabilities.

The techno-economic analysis of four scale-up strategies, including standard, prime, hybrid, and constrained, has been performed. When combined, our findings explicitly show the trade-off between centralization and modularity. At small system-module sizes (up to ~300 kW), centralized layouts are more economical because CAPEX from multiple small, low-TRL modules dominates. As scale increases, the hybrid modular layout becomes preferable, higher electrical efficiency and resource savings drive a lower LCOE (down to 0.155 $/kWh), since OPEX dominance amplifies efficiency gains. Thus, the optimal design strategy depends on three levers: (i) scale (module size and production volume), (ii) fuel price (and utility costs), and (iii) component maturity (learning curves for stacks and reformers). In practice, a mixed approach is likely, centralizing high-TRL BoP while modularizing stacks and reformers, evolving toward greater standardization as volumes grow.

In summary, we have addressed a critical challenge in SOFC scale-up by introducing a modular, efficient, and economically viable system layout. This approach supports flexible manufacturing, optimized component integration, and practical deployment at larger scales. We believe our findings provide a clear road map for advancing scalable SOFC technologies in support of global low-carbon energy transitions.

### Limitations and future work

While this study provides a comprehensive system-level and techno-economic assessment of modular SOFC scale-up, several limitations must be acknowledged. The present analysis is based on steady-state conditions and does not capture long-term degradation or transient behavior under dynamic loading. In reality, key components, particularly stacks, reformers, and heat exchangers, experience gradual performance decay due to thermal cycling, carbon deposition, and material fatigue, which can affect both efficiency and maintenance cost over time. Future work can incorporate degradation models to estimate lifetime efficiency trajectories and replacement costs, enabling a complete life-cycle cost assessment.

Another aspect requiring further development is system robustness under variable operating conditions. Dynamic simulations may be conducted to evaluate system response to changes in fuel composition, pressure drop, ambient conditions, and start-up/shutdown cycles, thereby improving understanding of operational resilience. These aspects were not covered comprehensively in the present work because the current study serves as a foundational step to highlight the importance of long-term energy storage scale-up strategies. We aim to raise awareness of the trade-offs between standardization and customization, encouraging collaborative efforts within the energy system community. Addressing these topics will require extensive experimental validation to confirm module-level performance and reliability under realistic conditions.

Finally, the proposed modular concept can be extended to reversible operation (solid oxide electrolyzer mode) to explore power-to-fuel integration and seasonal storage. The ultimate goal is to design dual-mode modules with integrated piping and valve systems that enable operation in both fuel-cell and electrolyzer modes. This represents another promising and technically demanding research direction that warrants dedicated investigation.

Overall, these future developments will provide a more complete understanding of long-term system viability and guide the industrial implementation of modular reversible solid oxide (rSOC) systems.

## Insights on scale-up strategy

Although this study focuses on SOFC systems, the stepwise approach we propose offers generalizable guidance for scaling up large-scale energy systems, an important but underexplored research question. Here, we address the scale-up challenge from three complementary perspectives. First, we explored a range of system configurations without fixing the number or size of various TRL component modules. This allowed for the evaluation of different pipeline layouts, series and parallel connections, and levels of system complexity, resulting in practical design guidelines for engineers working under technical constraints such as allowable pressure drops or component availability. Second, a case study based on a fixed-size system was conducted to assess the performance advantages of the proposed hybrid layout. The results confirmed the improvement in system efficiency and less reliance on external resources compared to conventional system designs. Third, we performed a techno-economic analysis by varying the system module size while maintaining a fixed global product requirement. This analysis identified which components are best centralized or decentralized to minimize the LCOE across different scaling scenarios.

From these three perspectives, several key insights emerged. Enhancing the performance of small-scale systems is essential prior to scale-up, as efficiency becomes increasingly critical when operating costs dominate. Although the economic evaluation does not account for all possible system complexities, the comparative LCOE analysis remains robust and demonstrates the importance of aligning market demand with manufacturing capabilities. Finally, modular systems provide greater operational resilience: whereas centralized systems may require full shutdown in the event of component failure, modular configurations allow for independent monitoring and localized intervention, ensuring higher system reliability and flexibility.

Together, these findings highlight the technical and economic feasibility of a standardized, modular approach to system scale-up, offering a practical foundation for both researchers and industrial stakeholders.

# Methods

## Modularity concept

Conventional design analysis of SOFC-based systems typically involves the development of process flow diagrams followed by detailed system integration and optimization tailored to specific industrial requirements. While this approach ensures design precision, it often results in diverse layouts that pose challenges for standardization and scalability. To address these limitations, this study proposes a generalized modular design framework that divides the system module into three primary groups: upstream, main, and downstream component modules.

As illustrated in Fig. 7a, the upstream modules include components for input preparation and handling, such as fuel and water tanks, a gas cleaning system, a water purification unit, and the electrical heater for start-up. The main component modules encompass critical power generation components, namely the reformer and the stack, and pressure and heat management components, namely pumps, compressors, blowers, and heat exchangers. Lastly, the downstream modules focus on post-processing and product management, comprising the catalytic burner, $CO_2$-water separator, and product and buffer tanks. Each component module is represented as a "box" of standardized sizes with fixed internal pipelines and layouts. Connections between component modules are facilitated through plug-and-play mixers and splitters, enabling straightforward system module assembly and disassembly.

By standardizing module types and sizes within each category, the proposed framework enables mass production of component modules, enhancing manufacturing efficiency and reducing costs. This modular approach shifts the focus from custom system design toward the development of standardized component modules that can be flexibly assembled into various system configurations. The capacity of the system module is directly determined by the number of stack modules, making scalability straightforward and adaptable to different electricity demand levels. The key challenge of this approach lies in determining the optimal sizes of component modules and establishing design principles for pipeline layout and system assembly, enabling the development of highly efficient power plants that comply with industrial constraints.

## Industrial constraints and design targets

While various aspects, such as cost and environmental performance of the SOFC system, can be improved, efficiency remains a critical metric as it reflects the fundamental technological limitations of the system design. The SOFC system has the potential to achieve high electrical efficiency through design and optimization; however, optimized results often overestimate what is feasible in industrial settings. To ensure the practical applicability of the system design, this study incorporates several critical and realistic industrial constraints, bridging the gap between theoretical design and a scaled-up industrial facility.

**Stack material degradation.** Carbon deposition can occur when hydrocarbon fuels, such as methane, are insufficiently reformed or when the stack operates under conditions favorable to carbon formation, resulting in solid carbon accumulation on the anode side. This leads to deactivation of the nickel-based catalyst, mechanical damage such as cracking or delamination, and a decline in electrochemical performance. Consequently, system efficiency is reduced, fuel consumption increases, and maintenance demands rise, ultimately elevating operational costs.

Several strategies can be employed to mitigate carbon deposition. Optimizing the S/C ratio and the reforming ratio in the pre-reformer ensures that internal reforming reactions, under appropriate operating conditions within the stack, effectively suppress carbon formation. Additionally, maintaining a sufficient level of unconverted fuel downstream of the stack prevents localized fuel starvation, which could otherwise promote carbon-forming reactions such as methane cracking and the Boudouard reaction. Therefore, in this study, several design specifications have been adopted based on the recommendations from the stack manufacturer[33,70]. The S/C ratio at the inlet of the stack is maintained above 1.5, the maximum internal reforming ratio within the stack is limited to less than 90% (single-pass fuel utilization), and more than 10% of the fuel must remain unconverted downstream of the stack.

**Stack performance improvements.** Efficient control of the S/C ratio effectively prevents carbon deposition but requires specialized equipment for external water purification, increasing the LCOE[71–74]. Additionally, the use of more external water requires significant heat input for evaporation, penalizing waste heat recovery and valorization.

AOG recirculation from the stack offers a potential solution to reduce external water consumption. AOG primarily contains carbon dioxide, water vapor, carbon monoxide, hydrogen, and methane. Recirculating a portion of the AOG increases global fuel utilization and reduces the need for fresh fuel and external water to maintain the S/C ratio in both the pre-reformer and the stack, thereby enhancing overall system efficiency. In conventional designs, a low-temperature AOG blower with a mechanical efficiency of approximately 0.8 is typically used for recirculation. Due to the temperature limitation of commercially available blowers, around 280 °C[28], the recirculated AOG must be cooled before entering the blower. The cooled AOG is then mixed with fresh fuel and steam and reheated to the pre-reformer inlet temperature. This approach presents several challenges, including heat losses during cooling and reheating, increased system complexity, and

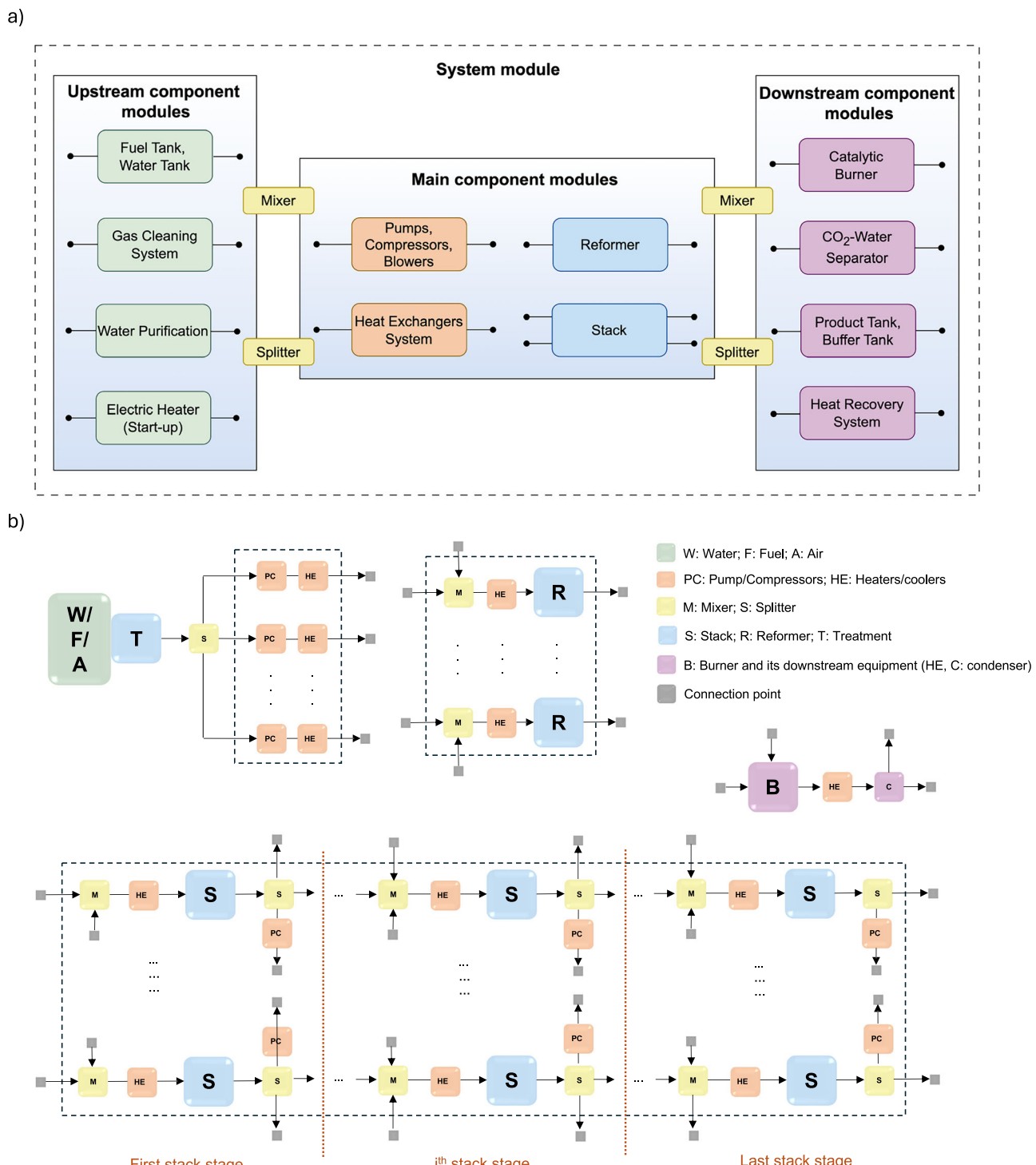

**Fig. 7 | Modular framework and hybrid layout of the solid oxide fuel cell (SOFC) power generation system. a** The modular framework of the SOFC power generation system with upstream, main, and downstream module groups; **b** modular layout of hybrid system design for SOFC scale-up: resource flow and component configuration.

limited efficiency gains in stacks with high single-pass fuel utilization, where the blower's power consumption may outweigh the benefits of the reduced external fuel input. Although high-temperature AOG recirculation using the high-temperature blower or customized ejector is a potential alternative, it offers limited advantages due to the low efficiency of high-temperature blowers and the reliance on non-standardized ejectors, which complicates process control and limits scalability. Therefore, when determining recirculation strategies for multiple stacks in large-scale systems, the evaluation should consider the benefits of AOG flow, such as its high steam content, which reduces

external steam requirements for maintaining the S/C ratio, and its contribution to improving global fuel utilization. At the same time, recirculation strategies must be carefully designed to avoid negative impacts on system performance, including reduced electrical efficiency due to AOG blower power consumption, excessive flow rates that increase pressure drops in the reformer and stack, potentially approaching design limits, and additional heat exchanger requirements or thermal losses.

Moreover, the application of COG recirculation also offers several benefits for SOFC systems, including reduced fresh air flow, lower

fresh air heater duty, and decreased power consumption by the fresh air blower that typically accounts for a 2–3% reduction in the system's electrical efficiency[28]. In the conventional system design, the unconverted fuel is combusted in a burner, and exhaust gases serve as the primary heat source to heat up the fresh air to the required stack inlet temperature. Reducing fresh air flow allows more heat from the burner to be available for the waste heat recovery system. However, COG recirculation can adversely affect stack performance by lowering the oxygen partial pressure on the cathode side, which reduces the single-pass conversion efficiency of the stack[75]. Thus, high system efficiency is typically achieved with limited COG recirculation, balancing its benefits and drawbacks. Thus, this study has considered these trade-offs in the development of efficient and scalable SOFC systems.

Finally, SOFC systems are increasingly recognized as a promising technology for combined heat and power applications[53,76]. Integration with micro gas turbines has shown significant improvement in electrical efficiencies[77]. These benefits are even more pronounced in large-scale SOFC systems, where waste heat from multiple stacks can be consolidated and recovered using larger, commercially available gas turbines. Such turbines are not only more cost-effective but also well-developed for efficient waste heat utilization. In this study, the waste heat from all stacks is analyzed collectively through a centralized heat recovery module.

**Pressure drops estimation.** In literature, pressure drops are typically considered based on industrial experience, which, while generally accurate, may lack robust scientific validation. In this study, rather than relying on such assumptions, scientific calculations are performed to compute pressure drops for key component modules, namely the stack, reformer, burner, and heat exchangers. These component modules are critical in determining the required pressure increase, which directly influences system electrical efficiency. By employing this approach, the study aims to provide a more reliable and scientifically grounded upper bound of pressure drop and its impact on the performance of large-scale systems. Pressure drops across various system components are systematically computed using a combination of stochastic simulations and fixed estimates.

Pressure drop computation for the stack is complex due to various interacting factors, as shown in Eq. (3), which incorporates fluid dynamics and material properties across the gas diffusion layer (GDL) of the cathode and anode[28]. Key parameters include flow rate, fluid density, and viscosity. The coefficient $C_{GDL}$ reflects the resistive properties of the GDL, determined by its material and structure, while the mass flow rate ($\dot{m}$) represents the fluid moving through the GDL. Adjustments for density $\left(\frac{\rho_{ref}}{\rho}\right)$ and viscosity $\left(\frac{\mu}{\mu_{ref}}\right)$ account for changes in compressibility and environmental conditions. These parameters collectively enable the calculation of the dynamic pressure drops, with reference values provided by the stack manufacturer.

$$\Delta P_S = C_{GDL} \cdot \dot{m} \cdot \frac{\rho_{ref}}{\rho} \cdot \frac{\mu}{\mu_{ref}} \tag{3}$$

For the pre-reformer, the pressure drop is determined through Monte Carlo simulations by sampling gas hourly space velocity (GHSV) and geometric parameters, specifically the length-to-diameter (L/D) ratio of the flow channel, from predefined uniformly distributed ranges. This approach captures diverse operational conditions and flow geometries. The resulting $\Delta P_R$ values are calculated using the Ergun equation, which describes fluid flow through packed beds (Eq. 4). Key parameters include reactor length ($L$), particle diameter ($d_p$), bed void fraction ($\varepsilon$), dynamic viscosity ($\mu$), fluid density ($\rho$), and superficial flow velocity ($v$) which is calculated using GHSV and reactor geometry. The particle diameter (3 mm) and corresponding void fraction (0.85) were

calculated from literature[78], representing a relatively free flow, typical for 15% cracking of $CH_4$ in the pre-reformer. The number of tubes (48) is also specified. Thermodynamic properties are retrieved from Aspen Plus simulations. More details on this calculation can be found in the SI, Section A1.1.

$$\Delta P_R = \frac{L}{d_p} \cdot \left( \frac{150 \cdot (1 - \varepsilon)^2 \cdot \mu \cdot v}{\varepsilon^3 \cdot d_p} + \frac{1.75 \cdot (1 - \varepsilon) \cdot \rho \cdot v^2}{\varepsilon^3} \right) \tag{4}$$

Similar to the pre-reformer, the burner pressure drop is estimated using Monte Carlo simulations by uniformly sampling GHSV and the L/D ratio. In addition, the pressure loss coefficient ($K_{sing}$), which accounts for singular losses due to abrupt changes in geometry (e.g., gas train, nozzle, and sub-components), is sampled from a uniform distribution. Unlike the pre-reformer, the burner lacks catalysts and operates under different flow conditions; therefore, the pressure drop is calculated using the Darcy-Weisbach equation to account for both frictional and singular losses (Eq. 5). The friction factor ($f$) is determined from the Reynolds number, fluid density ($\rho$) is obtained from Aspen Plus simulations, and flow velocity ($v$) is derived from GHSV and burner geometry. Due to model simplifications and parameter uncertainties, a safety factor of 3 is applied. Pressure drops across heat exchangers, and condensers are assumed to be fixed at 10 mbar each. The obtained pressure drops for all components are in line with other studies[79,80]. More details on this calculation can be found in the SI, Section A1.2.

$$\Delta P_B = f \cdot \frac{L}{D} \cdot \frac{\rho \cdot v^2}{2} + K_{sing} \cdot \frac{\rho \cdot v^2}{2} \tag{5}$$

### Hybrid system modular design

**Model parameters.** Although this work is a system-integration and techno-economic study rather than detailed experiment testings, we have employed a steady-state SOFC system (module) model whose parameters are taken from manufacturer specifications, European projects, and prior experimental studies[51,55,57,81–84].

Moreover, it is important to point out that this study does not aim to re-validate SOFC system models. We acknowledge that most of the published validation studies are not at the hundreds-of-kilowatt scale; however, this does not preclude scale-up. Whole-plant validation at very large sizes is rarely practical and, by definition, departs from the modularity concept that underpins industrial deployment and experimental prototyping. Instead, we adopt a module-level strategy: standardized modules are validated across multiple scales, and the steady-state behavior of critical components is observed to be consistent within established operating envelopes. Consequently, although full-system validation at a very large scale remains an uncertainty, we judge its impact on the conclusions here to be limited, as our results hinge primarily on architectural trade-offs and trend robustness rather than absolute point estimates.

In the case study (subsection "Case study on a 50 kW SOFC system"), each stack module is modeled as a 10 kW unit with an approximate stack area of 3 m², reflecting the most common size of commercially available single-stack modules. The current density is set to 0.4 A/cm², with a maximum internal reforming ratio of 90%. Stack inlet and outlet temperatures are fixed at 680 and 750 °C, respectively, in accordance with manufacturer specifications[28]. Additionally, the mechanical efficiency of the low-temperature blower is set to 0.8, and the burner's maximum operating temperature is limited to 900 °C, based on the cost-efficient material's thermal resistance limit[76].

**Layout optimization.** Figure 7b illustrates the concept of scaling up the SOFC system through a modular framework aimed at optimizing resource flows and reducing system complexity. The process begins with centralized storage of treated water and fuel, which are distributed via pumps, compressors, and heaters in a decentralized

parallel configuration before entering the reformer section. Here, layout optimization determines the optimal number and configuration of standardized reformer modules. Three reformer-stack connection types are considered: one-to-one, one-to-many, and many-to-one. The optimization identifies the most efficient arrangement and sizing to support system scalability and performance.

The processed flow from the reformer section is directed to the stack section, which consists of stack modules arranged in either series or parallel configurations. Within each stack stage (e.g., $i$th stage), the anode side includes a mixer and heater upstream of the stack and a splitter downstream. The mixer combines incoming streams from the reformer, previous stack stages, or recycled flows from other stack stages within the series-parallel configuration, while the splitter distributes the outlet stream to subsequent stack stages, the burner section, or other stack stages as needed. In case of recirculation from a downstream stage ($i$th) to the upstream stage ($[i-1]$th), the flow passes through a blower and/or a heat management unit to adjust the temperature and pressure before entering the mixer.

Moreover, the stack section layout in Fig. 7b illustrates both the anode- and cathode-side flow configurations. Air is supplied to the cathode to facilitate electrochemical reactions and regulate the stack outlet temperature. As with the water and fuel sections, the air supply system employs blowers arranged in a decentralized parallel configuration. Effective airflow management ensures uniform thermal conditions across all stack modules, which is critical for stable and efficient operation.

As shown in Fig. 7b, the downstream flow from the anode side of the stack modules can be directed to the burner and its downstream component modules introduced in Fig. 7a (burner, heat exchangers, condenser, etc.). The system may have either centralized or multiple burners, depending on the available commercial burner sizes. The primary objective is to minimize the unconverted fuel reaching the burner, thereby maximizing global fuel utilization. In this study, oxy-combustion is considered[28]. After cooling, the flow is directed to centralized or multiple $CO_2$-water separator modules. $CO_2$ or water can be recycled back to the burner to control the combustion temperature, preventing it from exceeding the material temperature limits. Additionally, the burner's output stream can be integrated with downstream modules, such as a centralized Rankine cycle, to enhance waste heat recovery and valorization.

Overall, the primary goal of this scale-up design is to optimize the plant layout by minimizing the consumption of water, fuel, and air, as well as reducing the number of component modules while maximizing the overall system's electrical and thermal efficiencies. The optimization process also determines the optimal configuration of key component modules, such as reformers and stacks, specifying the numbers and sizes of reformers and the arrangement of stacks in a series-parallel configuration. By solving this optimization problem, the proposed design achieves an effective balance between scalability, resource efficiency, and cost-effectiveness.

**Operating conditions optimization.** After determining the optimal system configuration, this study further considers an additional layer of optimization focused on the operating conditions of individual component modules. These modules are not required to operate under identical conditions, as parameters such as reformer temperature, external reforming ratio, and the input quantities of fresh fuel and water may vary across the system. To address this, a multi-objective optimization is conducted with the goals of maximizing electrical efficiency and global fuel utilization while minimizing external water consumption, as summarized in the SI, Table A3.1. The optimization is subject to industrial constraints, including a minimum S/C ratio of 1.5 before the stack, a requirement that at least 10% of the fuel remains unconverted downstream of the stack, and a maximum allowable pressure drop of 100 mbar across both the anode and cathode sides.

This approach underscores the flexibility of the modular design, enabling optimized system operation within practical industrial limits[33,82].

**Techno-economic analysis**

To evaluate and compare different SOFC scale-up strategies, it is also crucial to introduce economic performance indicators. A central objective of this study is to explore how the trade-off between centralization and decentralization affects complexity, cost, and scalability. This involves analyzing the relationship among the global electricity output (i.e., total electricity generated from the stack modules of all system modules, defined in Fig. 7a), the number of component/system modules, and the size of component/system modules. This study assumes a fixed global electricity output of 1 GW ($P_{\text{global}}$), representing the demand for a large urban area or small regional grid. This output is supplied by deploying multiple identical system modules, each contributing a portion of the total output such that their combined capacity equals 1 GW.

**Overview of scale-up strategies.** A key design variable is the system module size ($P_{\text{system}}$), defined as the total electricity generated by all stack modules within a single system module. As illustrated in Fig. 7, each system module integrates the necessary component modules, such as stacks, reformers, burners, and BoP modules, into a fully operational SOFC unit. Notably, each system module may include multiple component modules of the same type, depending on the size availability and scale-up strategy adopted. To investigate a wide range of configurations, this study includes system module sizes varying from 10 kW to 1 GW.

A distinguishing feature of this techno-economic analysis is the explicit consideration of cost behavior for low TRL components, specifically stack and reformer modules. Unlike high-TRL components (e.g., heat exchangers, blowers), which benefit from globally standardized production across multiple sectors, low-TRL components are unique to SOFC systems and exhibit two counteracting cost trends: economies of scale, which reduce unit cost with an increase in unit size, and production volume effects, which increases unit cost with a decrease in the number of manufactured units. To capture this distinction, the number of component modules is categorized as $N_{\text{low-TRL}}^X$ and $N_{\text{high-TRL}}^X$, where superscript $X$ denotes the specific system design strategy under consideration.

This study analyses four scale-up strategies. The first strategy, referred to as the standard system design strategy A, assumes that all component modules within a system module are centralized. That is, each system module contains one stack, one reformer, one burner, and one set of BoP components. To meet the 1 GW global electricity output, the number of system modules is determined by dividing $P_{\text{global}}$ by $P_{\text{system}}$. Consequently, the number of low-TRL and high-TRL modules in strategy A are the same, as given by Eq. (6). In this configuration, each component module is sized to match the capacity of the system module, as shown in Eq. (7). Here, $P_{\text{stack}}^A$ and $P_{\text{reformer}}^A$ represent the capacities of a single stack and reformer modules under strategy A, while $P_{\text{rest}}^A$ denotes the capacity of each high-TRL component module (e.g., burner or heat exchanger). Stack capacity reflects electrical power output, reformer capacity corresponds to fuel processing throughput, and high-TRL capacities represent thermal or flow requirements at the system module scale.

$$N_{\text{low-TRL}}^A = N_{\text{high-TRL}}^A = \frac{P_{\text{global}}}{P_{\text{system}}} \tag{6}$$

$$P^A_{\text{stack}} = P^A_{\text{reformer}} = P^A_{\text{rest}} = P_{\text{system}} \qquad (7)$$

The second strategy, referred to as the hybrid system design strategy B, corresponds to the layout introduced in this study. In this configuration, each system module contains $(L + K)$ stack modules, $K$ numbers operating in parallel and $L$ numbers in series, along with $(L + K)$ reformer modules. However, the remaining BoP components are centralized per function (e.g., one burner and one air) within each system module. Thus, while the number of high-TRL component modules remains unchanged from strategy A, as shown in Eq. (8), the number of low-TRL component modules is $(L + K)$ times greater. Each high-TRL component module is sized to serve the full capacity of the system module, whereas the stack and reformer modules are each sized to handle one-$(L + K)$th of the system module capacity, as defined in Eq. (9). From the results, it is observed that $L + K = 5$ for the 50 kW system used as an example, as shown in the equations. The rationale behind this value is explained in the subsection "Case study on a 50 kW SOFC system."

$$N^B_{\text{low-TRL}} = (L + K) \times N^B_{\text{high-TRL}} = (L + K) \times \frac{P_{\text{global}}}{P_{\text{system}}} ; L + K = 5 \qquad (8)$$

$$(L + K) \times P^B_{\text{stack}} = (L + K) \times P^B_{\text{reformer}} = P^B_{\text{rest}} = P_{\text{system}} ; L + K = 5 \qquad (9)$$

The third strategy, referred to as the constrained system design strategy C, builds upon the concept established in strategy B. In this configuration, the capacities of each stack and reformer module remain one-$(L + K)$th (one-fifth) of the system module capacity. However, unlike strategy B, this approach assumes that all high-TRL component modules (i.e., BoP components) are also decentralized in each system module. That is, they follow the same number and sizing logic as the low-TRL component modules, as shown in Eqs. (10) and (11). This strategy is introduced to evaluate the economic performance of fully decentralized system modules with respect to the hybrid system design strategy B. It quantifies the cost implications of applying a uniform decentralization approach across all component modules within a system module.

$$N^C_{\text{low-TRL}} = N^C_{\text{high-TRL}} = (L + K) \times \frac{P_{\text{global}}}{P_{\text{system}}} ; L + K = 5 \qquad (10)$$

$$(L + K) \times P^C_{\text{stack}} = (L + K) \times P^C_{\text{reformer}} = (L + K) \times P^C_{\text{rest}} = P_{\text{system}} ; L + K = 5 \qquad (11)$$

The fourth strategy is referred to as prime system design strategy D, which builds upon the framework of strategy A but introduces a further degree of centralization. In this configuration, to meet the fixed global electricity output, the stack and reformer modules (low-TRL components) are still in the same manner as in strategy A. However, all other high-TRL BoP component modules are fully centralized and deployed only once to serve the entire global electricity needed. As a result, the number of low-TRL components is still calculated as the ratio of the global electricity output to the system module capacity, as shown in Eq. (12), whereas the number of high-TRL components is fixed at one with 1 GW capacity, as indicated in Eq. (13).

Regarding sizing, stack and reformer modules remain matched to the capacity of a single system module, as in strategy A (Eq. 14). In contrast, the capacity of the centralized high-TRL BoP modules must scale with the entire global electricity output, as given in Eq. (15). This approach represents an idealized lower bound in terms of BoP centralization. While this strategy may not be entirely feasible in real-world applications due to equipment sizing limitations or process integration challenges, it serves as a useful benchmark for evaluating the

techno-economic trade-offs associated with decentralization versus centralization.

$$N^D_{\text{low-TRL}} = \frac{P_{\text{global}}}{P_{\text{system}}} \qquad (12)$$

$$N^D_{\text{high-TRL}} = 1 \qquad (13)$$

$$P^D_{\text{stack}} = P^D_{\text{reformer}} = P_{\text{system}} \qquad (14)$$

$$P^D_{\text{rest}} = P_{\text{global}} \qquad (15)$$

**Capital cost estimation.** Due to the limited availability of cost data for different component modules across scales, this analysis utilizes information from publicly available resources[78,85]. The cost calculation methods and equations are implemented in Python, which will be available to interested readers. Therefore, detailed cost calculations are not presented; only cost calculation basics are discussed below.

The manufacturing cost of each component module is estimated using a log-log regression, which captures the combined effects of component scale and production volume. This relationship is expressed in Eq. (16). In this equation, $P$ represents the scale/size of the component, typically measured in terms of power capacity (e.g., kW), and $N$ denotes the annual production volume or the number of units produced per year. The coefficients $a$, $b$, $c$, $d$, and $e$ are empirical parameters fitted using available cost data derived from literature or industry benchmarks. The cost function includes a linear and quadratic logarithmic dependence on size ($\log(P)$ and $[\log(P)]^2$) to account for economies of scale, which tend to lower the levelized cost with an increase in the size of the component. The $\log(N) [\log(N)]^2$ terms reflect how increased production volume can reduce costs through manufacturing efficiencies and experience-based improvements. For high-TRL components such as pumps and compressors, the production volume is fixed at 10,000 units to represent a mature manufacturing level. Their costs are considered stable and largely independent of fuel cell-specific market dynamics. In contrast, for low-TRL technologies such as reformers and stacks, the production volume depends on the component capacity and the global electricity output. Fitting results alongside the exact cost correlations for each component module can be found in the SI, Section A2 (Figs. A2.1–A2.4, Table A2.5).

$$\log(C_{cp}) = a + b \cdot \log(P) + c \cdot [\log(P)]^2 + d \cdot \log(N) + e \cdot [\log(N)]^2 \qquad (16)$$

From the estimated component purchase (or manufacturing) cost ($C_{cp}$), the capital investment is calculated for each component module, accounting for component-specific lifetimes, 5 years for stacks and reformers, and 20 years for other high-TRL component modules. The component module investment cost (CAPEX$_{\text{cm}}$) is then calculated through Eq. (17), where $r$ is the number of replacements, $i$ is the interest rate, $t_c$ is the component lifetime, $t$ is the project lifetime.

$$CAPEX_{cm} = \sum_{r=0}^{r} \left( C_{cp} \cdot \frac{1}{(1+i)^{r \cdot t_c}} \right) + C_{cp} \cdot \frac{t - r \cdot t_c}{t_c} \cdot \frac{1}{(1+i)^{r \cdot t_c}} \qquad (17)$$

**Levelized cost of electricity.** The LCOE is a key metric for evaluating and comparing the cost-effectiveness of different design strategies. It accounts for annualized investment, operating and maintenance costs, and electricity generation. As shown in Eq. (18), $C_{\text{sm},a}$ represents the annualized system module investment cost calculated with interest rate ($i = 5\%$) and project lifetime ($t = 20$ years). The operating cost ($C_{\text{op}}$) is computed using fuel cost (FC), electrical efficiency ($\eta_{\text{elec}}$) of the

SOFC system, and total electricity output ($E_{elec,out}$) from the system for 8000 h/year (Eq. 19). Finally, the maintenance cost ($C_{maint}$) is defined as 5% of the annualized investment cost for the system module.

$$C_{sm,a} = \sum_{\text{All Component Modules}} CAPEX_{cm} \cdot \frac{r \cdot (1+i)^t}{(1+i)^t - 1} \qquad (18)$$

$$C_{op} = \frac{E_{elec,out}}{\eta_{elec}} \cdot FC \qquad (19)$$

$$LCOE = \frac{C_{sm,a} + C_{op} + C_{maint}}{E_{elec,out}} \qquad (20)$$

**Uncertainty analysis.** To account for parameter uncertainty and assess the robustness of the economic outcomes, an uncertainty analysis is also conducted. Capital cost values of low-TRL (stack and reformer) and high-TRL components (e.g., BoP modules) are varied based on the normal distribution with a standard deviation of 20 and 10%, respectively. Component lifetimes are uniformly sampled within specified ranges (5–12 years) for stacks and reformers, and the normal distribution with a standard deviation of 10% is applied to the lifetimes of high-TRL components. The discount rate is also varied with a standard deviation of 10%, while the fuel price and the project lifetime are kept constant at 10 cents/kWh and 20 years, respectively. The uncertainty analysis is performed over 1000 times to ensure statistical robustness. More information on the parameter uncertainty can be found in the SI, Table A2.6.

## Data availability

The data supporting the findings of this study are available in the paper and its Supplementary Information. Source data are provided with this paper.

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

## Acknowledgements

The author (X.W.) thankfully acknowledges the financial support from the project AMON. The project is supported by the Clean Hydrogen Partnership and its members under grant agreement no. 101101521. Co-funded by the European Union. Views and opinions expressed are, however, those of the author(s) only and do not necessarily reflect those of the European Union or the Clean Hydrogen Partnership. Neither the European Union nor the granting authority can be held responsible for them. The author (X.W.) also thankfully acknowledges the financial support from the project H2Marine. The project is supported by the Clean Hydrogen Partnership and its members under grant agreement no. 101137965. Co-funded by the European Union. Views and opinions expressed are, however, those of the author(s) only and do not necessarily reflect those of the European Union or the Clean Hydrogen Partnership. Neither the European Union nor the granting authority can be held responsible for them. This work was supported by the Swiss State Secretariat for Education, Research, and Innovation (SERI) under contract number 24.00109 - 101137965.

## Author contributions

Conceptualization: X.W., S.S., F.M. System modeling and optimization: X.W., S.S. Economic analysis: A.W., X.W., S.S. Writing—original draft: X.W., S.S., A.W. Study revision: X.W., S.S., H.Y., A.W. Supervision, review: F.M., J.V.h.

## Competing interests

The authors declare no competing interests.
