## [Transparent Peer Review file · Nature Communications]

Scalable Modular Design of Solid Oxide Fuel Cell Systems for Enhanced Large-Scale Power Generation

Corresponding Author: Dr Xinyi Wei

Version 0:

Reviewer comments:

Reviewer #1

(Remarks to the Author)

This study introduces a design framework for large-scale SOFC systems, which is extremely important for SOFC technology but has been rarely discussed in the literature. Through a comprehensive techno-economic analysis, the study compares the costs associated with various scaling-up strategies. This comparison is of paramount importance for the application of SOFC systems, given that the most significant limitation of SOFC technology is the relatively low rated power of individual SOFC stacks. I particularly appreciate the comparison and discussion of the advantages and disadvantages of centralization and decentralization. However, I do have several concerns. Many details and motivations are lacking in the study, and the methodology employed is rather complex, which makes it somewhat challenging to fully grasp and evaluate.

1. The design approach under discussion centers on a highly specialized SOFC system configuration that incorporates AOG and COG recirculation. I am questioning the necessity of off-gas recirculation in this context. As illustrated in Table 1, the NCNA design, which does not feature off-gas recirculation, exhibits an efficiency that is only 0.7% lower than that of the proposed design with off-gas recirculation. Moreover, the NCNA design is significantly more straightforward. The author should answer which design is more commonly utilized in existing SOFC systems.

2. Regarding the proposed multi-module system, the author presents a hybrid design where two stacks are first connected in parallel, followed by three stacks connected in series. For simplicity, I refer to this design as "2-1-1-1". I have several questions on this matter.

Firstly, why was the "2-1-1-1" design chosen over alternatives such as "2-2-1", "3-1-1", "4-1" etc.? The author should provide the rationale and motivation behind this particular design.

Secondly, I am concerned about the practicality of this design. Generally, system flexibility decreases as internal coupling increases. The hybrid design significantly raises the system's coupling level because the operating conditions of downstream modules are contingent upon those of upstream modules.

Thirdly, as the author mentioned, increasing pressure for upstream stacks may result in gas leakage, which poses a serious safety risk. As far as I know, the SOFC stack fabricated by Elcogen can only endure a pressure difference of 50 mbar. Are there any actual SOFC systems that employ the serial connection of multiple stacks?

Lastly, current large SOFC systems typically integrate multiple stacks within a single hotbox, with stacks connected in parallel in terms of gas flow. I am uncertain whether the stacks in the author's design are all integrated within the same hotbox or if each stack has its own independent hotbox.

3. I question the necessity and reasonability of employing Monte-Carlo simulation to model the pressure drops of the reformer and burner, given the absence of actual random variables. In Section B2, the assumption that GHSV and geometric parameters follow a normal distribution appears perplexing to me. These parameters are not truly "random" in nature. If the intention is to perform sensitivity analysis, I would suggest that a uniform distribution seems a more rational choice.

4. On page 6, line 124, it states, "Overall, optimum AOG usage within the hybrid system design for scaling up involves two stacks operating in parallel." I'm curious about the rationale behind choosing two as the optimal number of parallel stacks. What led to this specific configuration rather than any other number of stacks operating in parallel?

5. Page 7, line 153, why did the stacks operating in series have higher pressure drops? Was it caused by the difference in flowrates?

6. On page 2, line 32, it is mentioned that SOFCs have demonstrated minimal degradation. I don't quite agree with this statement. In my view, degradation and high maintenance cost continue to be a critical issue for the application of SOFCs. The satisfactory lifetimes demonstrated in existing studies are mostly achieved with single cells (even button cells) or short stacks under laboratory conditions. These results may not be directly applicable to real industrial settings.

7. In Figure 3, you mentioned the heat exchanger module, but I didn't see any heat exchangers in Figure 3 and Figure 4.

8. In Figure 4, the terms “10kW stack” and “50kW stack” are mentioned. The authors should clarify whether these terms refer to a single stack or to a stack module that consists of multiple stacks integrated within a hotbox. Additionally, in Figure 3, it is unclear whether the parallel stack modules are located in two independent hotboxes or within the same hotbox.
9. On page 15, line 336, it states, “...we have presented a comparative techno-economic analysis of the four scale-up strategies for SOFC system, introduced earlier...”. I found this statement confusing because I couldn't locate any prior introduction to the scale-up strategies. It was only after thoroughly reading through the paper that I discovered the introduction was actually in Section 4.4.1 on page 30.
10. Page 26, line 601, you mentioned “more than 10% of the fuel must remain unconverted downstream of the stack”. However, in line 609, you mentioned “AOG primarily contains carbon dioxide, water vapour, and trace amounts of carbon monoxide, hydrogen, and methane.” I think these two sentences are contradictory.
11. Page 30, line 756, “As illustrated in Figure ??” It appears that there is an issue with the cross-referencing.
12. Figure 5 illustrates the relationship between LCOE and module size for different system designs. Given that the maximum SOFC module size currently only reaches 100 kW, I believe the standard and prime designs are already sufficient. Moreover, I consider it impractical to construct a 1 GW SOFC module.
13. In Figure 3, the authors should clarify the meanings of ‘1X’, ‘2X’, and ‘3X’.

Reviewer #2

(Remarks to the Author)

In this manuscript, the authors present a modular hybrid design strategy for scaling up SOFC systems, incorporating both anode off-gas and cathode off-gas recirculation. The case study of a 50 kW SOFC system is detailed and demonstrates that the electrical efficiency can reach 66.3%. The design also achieves a reduction in external water and fresh air intake by 60% and 22%, respectively, compared to a baseline system. Moreover, the manuscript includes a techno-economic analysis of four scale-up strategies, showing that the hybrid approach results in the lowest levelized cost of electricity (0.155 \$/kWh) at the 1 GW scale, making it particularly suitable for large-scale deployment.

The system design is comprehensive and systematic, offering a well-structured strategy for modular SOFC integration.

Overall, this study is robust and highly relevant to the fields of energy, energy conversion, and environmental science. I believe it is suitable for publication in Nature Communications, pending minor revisions as outlined below:

1. While the manuscript presents a well-developed system design and techno-economic analysis, it would benefit from a brief discussion of limitations related to long-term operation. In particular, a quantitative or qualitative treatment of component degradation and lifecycle performance, especially for stacks and reformers, could provide a more complete picture of system viability over time. Including a short limitations paragraph addressing these factors, along with the underlying assumptions regarding component scalability and cost behavior, would help contextualize the results and guide future work.
2. The paper offers a solid comparative analysis of scale-up strategies, showing how different configurations perform across varying system sizes. However, the conclusion could more clearly highlight the trade-offs identified between centralized and modular approaches, particularly the shift in economic viability as system capacity increases. Emphasizing that the optimal strategy depends on scale, fuel price, and component maturity would help reinforce the broader applicability of the findings and underscore the practical considerations involved in SOFC system design.
3. In line 75, the term “AOG” is introduced without definition. It is only defined later (line 608). For clarity, it should be defined at its first occurrence.
4. In Figure 6a, the bar plot showing normalized CAPEX contributions of component modules across scale-up strategies at the 1 GW scale includes values of 5%, 6%, 2%, and 2% for strategies C, B, A, and D, respectively. It is unclear which specific equipment or components these values refer to. Please revise the figure or caption for clarity.
5. Line 756 contains a broken or missing reference to a figure. This should be corrected.

Reviewer #3

(Remarks to the Author)

The author proposes a design that diverges from conventional approaches to recovering fuel exhaust from the SOFC anode side. In this design, the AOG gas is directed into a secondary set of SOFCs to improve overall fuel utilization. The study explores various parallel and series configurations and optimizes operating parameters to achieve higher energy efficiency and fuel utilization.

As this is a system-level simulation, the device-specific parameters are based on values from existing literature or commonly adopted industry standards. Empirical formulas are used to model subsystem details. In the latter part of the paper, a techno-economic analysis and an economic modeling exercise are presented.

The main claimed innovation—reusing tail gas via an additional SOFC unit arranged in parallel or series—is no longer novel at this stage. There are already numerous publications that have explored similar designs, often with more technical detail and stronger implementation strategies. For instance, this paper simplifies the system-level model by treating the stack pressure drop as a constant. However, such simplification undermines the model's generalizability, as pressure drops vary significantly with stack power capacity and internal flow field geometry. How can this model be scaled? The pressure loss will change entirely, rendering the original conclusions inapplicable.

Another issue with this cascade concept is the considerable increase in system complexity. While not inherently unacceptable, such complex systems demand rigorous system modeling to identify the optimal sizing and configuration—such as how the power ratio between the first-stage and second-stage SOFCs is determined. Is this ratio arbitrarily chosen, based on intuition, or derived through some optimization algorithm? How is self-sustainability of the system ensured, and

how does one avoid scenarios where part of the system falls into an energy-deficient state that requires external input?

The economic model is overly optimistic. The final projected electricity cost is not the result of system optimization but rather a consequence of assuming reduced CAPEX due to large-scale manufacturing. In reality, SOFC demand is unlikely to reach such scales.

I regret to issue a recommendation for rejection.

This judgment stems from a combination of factors: the cascade design is already well explored in the literature (but the dynamic operation with more modeling details can be a future direction, or the experimental optimization of such system as it's really challenge to achieve self-sustainable); a simple review would reveal numerous works offering more nuanced modeling or experimental verification. On top of this, the modeling details in the present work are insufficient, and the model's applicability to larger systems is questionable. It may only be valid for sub-kilowatt stacks, in which case, stack-level simplifications of this kind are not acceptable. Lastly, the economic model is too idealized, producing a highly attractive electricity price figure that is unrealistic—even coal-fired or gas turbine power plants cannot deliver power at \$0.155/kWh in current market conditions.

Reviewer #4

(Remarks to the Author)

The authors have addressed a very important and interesting topic in the upscaling of SOC systems. At the beginning of the article, they point out the reversibility of SOC technology, but ultimately focus more on the SOFC fuel cell. However, it would be more interesting to see how the design affects the SOEC. Both may require a different setup at the system level. In this case, the rSOC character would be ineffective. The authors do not address industrial interconnection sufficiently at the end – Bloom Energy is mentioned. Overall, the article is well-written. A weakness becomes apparent in the literature review: too few citations (41) for such an "old" and well-known research area. Furthermore, the article lacks inclusion of publications from important European and American companies, universities, and research institutions. In this respect, the article is highly disappointing. Given the quality of the journal, Nature Communications, I would not recommend this article for publication. The article reflects good engineering practice – but has not been sufficiently scientifically evaluated. It would certainly fit better in another journal.

Version 1:

Reviewer comments:

Reviewer #1

(Remarks to the Author)

The authors have addressed all the comments carefully. I thus suggest the acceptance of this manuscript.

Reviewer #2

(Remarks to the Author)

This paper is well revised according to reviewer's suggestions. Now I recommend its publication.

Reviewer #3

(Remarks to the Author)

The authors have provided detailed responses to my questions, supported by sufficient literature and sound logical reasoning. In light of their explanations, I find the assumptions and model settings in the manuscript to be much more reasonable. I therefore support the acceptance of this manuscript for publication in this journal.

Reviewer #4

(Remarks to the Author)

The authors have thoroughly revised their manuscript and addressed all points raised in the first round of review. The responses provided are clear, detailed, and demonstrate a careful engagement with the reviewers' critiques. The revisions have substantially improved the clarity, rigor, and overall coherence of the paper. All previously identified weaknesses have been adequately resolved. I therefore consider the manuscript suitable for publication in its current form.

Scalable Modular Design of Solid Oxide Fuel Cell Systems for Enhanced Large-Scale Power Generation

Xinyi Wei, Arthur Waeber, Shivom Sharma, Hangyu Yu, Jan Van herle, Francois Marechal
Corresponding authors: xinyi.wei@epfl.ch

Dear Editor and Reviewers:

First, we would like to express our gratitude for your dedicated time reviewing our manuscript. Your insightful and constructive feedback has been invaluable in enhancing the quality of our study and broadening its potential impact on Nature Communications' readers.

We have carefully revised the manuscript and addressed each point raised by the editor and reviewers. This document provides a detailed account of our response to each comment and the corresponding changes made to the manuscript. While the key findings and conclusions remain unchanged, consistent with the overall positive feedback from all reviewers. The revisions have substantially improved/added the key discussion, clarity, rigor, articulation of limitations, and perceived impact of the study. All page and line numbers in this document refer to the revised manuscript.

We are grateful for the opportunity to refine our work with your guidance and hope this revised version meets your expectations. Thank you once again for your time and consideration.

Sincerely Yours,

Xinyi Wei, Arthur Waeber, Shivom Sharma, Hangyu Yu, Jan Van Herle, Francois Marechal

Short Highlights to Reviewer 1:

Thank you very much for your thorough review and insightful suggestions. Your comments covered all key aspects of the manuscript, from the introduction and methodology to the presentation of results, and we have fully addressed them in the revised manuscript.

We would especially like to thank you for your careful reading and critical suggestions, particularly your questions regarding the rationale for the proposed hybrid layout compared with traditional designs, as well as your insightful remarks on the hotbox configuration and heat-exchanger module. These points prompted us to think more comprehensively about several aspects that were not fully discussed in the original manuscript, and we have now incorporated these considerations into the revision. We fully agree with your recommendations and have included the corresponding concepts, extended discussion, and supported references in the revised manuscript.

All detailed responses are provided below. Thank you again for your expert feedback and valuable support.

The following Table addresses 1st Revision comments from Reviewer 1.

No.	Comments	Corrections and Responses
1	The design approach under discussion centers on a highly specialized SOFC system and COG recirculation. I am questioning the necessity of off-gas recirculation in this context. As illustrated in Table 1, the NCNA design, which does not feature off-gas recirculation, exhibits an efficiency that is only 0.7% lower than that of the proposed design with off-gas recirculation. Moreover, the NCNA design is significantly more straightforward. The author should answer which design is more commonly utilized in existing SOFC systems.	Thank you for the valuable suggestion. Indeed, the straightforward and most deployed layout today is NCNA (no AOG, no COG), largely because the configuration is simpler and easier to control. However, NCNA faces several limitations. First, even small efficiency gains matter, especially when SOFC systems are paired with reversible operation for long-duration storage, as small percentage improvements compound over time. For this reason, several European demonstration projects (e.g., CH2P, SWITCH, Nautilus, BLAZE, AMON) have implemented either AOG or COG recirculation. Most designs adopt warm AOG recirculation: the AOG is cooled to ~280 °C and compressed with a blower (commercial blowers are typically limited to this temperature), then compressed flow is injected upstream of the reformer and reheated to the reformer inlet temperature. AOG recirculation can increase electrical efficiency and exploit the steam content of the off-gas, thereby reducing fresh-fuel demand and the heat otherwise required to generate external steam. Less fuel lowers cost; Less required heat frees more waste heat for valorization; and reduced external water demand allows for a smaller water-purification unit, which can be costly in many regions (on the order of 3 €/m³ water, depending on quality).

		That said, our analysis also shows that the commercial AOG blower can offset or even negate these efficiency benefits, an important practical limitation. This is precisely the issue our work addresses: how to retain the benefits of AOG recirculation while avoiding dependence on a blower. In our results, the efficiency gain (without a Rankine cycle) is modest, but the techno-economic analysis demonstrates that even a small improvement can materially impact cost at large scale. In addition, the proposed design reduces external water use and enables direct heating that shifts recoverable heat from high to medium temperature; the latter is more readily convertible in a Rankine cycle, yielding ~1.5% efficiency improvement with the Rankine cycle. For a fair comparison, we performed multi-objective optimization for both NCNA and the proposed design. The optimized NCNA case operates at an S/C ratio very close to 1.5, the lower bound, which has the best system performance but may increase long-term risk of carbon deposition. In contrast, the proposed layout, with AOG containing more steam, naturally operates at a higher S/C ratio, potentially reducing carbon-formation risk in the reformer. Quantifying that durability benefit requires dynamic modeling and long-term tests, which are outside the scope of this study; but we have added this point to the “Limitations and Future Work” section. We recognize that the preliminary version did not explain these points clearly. The revised manuscript now provides explicit reasoning and supporting discussion. Please see page 3, lines 40–53; page 18, lines 423–422; and page 26, lines 632–656; page 27, lines 671-685.
2	Regarding the proposed multi-module system, the author presents a hybrid design where two stacks are first connected in parallel, followed by three stacks connected in series. For simplicity, I refer to this design as “2-1-1-1”. I have several questions on this matter. Firstly, why was the “2-1-1-1” design chosen over alternatives such as “2-2-1”, “3-1-1”, “4-1” etc.? The author should provide rationale and motivation behind this particular design.	Thank you for the insightful comments/questions. Rationale for the “2–1–1–1” configuration: We appreciate the opportunity to clarify the reasoning behind this design choice. The main motivation is to maximize the utilization of high-temperature heat through direct mixing. Our goal is to harness (thermal and chemical) energy of the AOG not only for unconverted fuel recovery but also for preheating the inlet flows to the reformer and stack inlets, thereby minimizing heat loss to the burner. To achieve thermal balance, the total flow from two parallel stacks ($2 \times X$ kW) represents the minimum requirement for providing the necessary preheating duty to the subsequent reformer and stack. Further, this configuration can optimally supply steam from an upstream stage to the subsequent downstream stage. While alternative configurations such as 3–1–1 or 2–2–1 are technically feasible, they would lead to heat imbalance, with surplus flow

	Secondly, I am concerned about the practicality of this design. Generally, system flexibility decreases as internal coupling increases. The hybrid design significantly raises the system's coupling level because the operating conditions of downstream modules are contingent upon those of upstream modules. Thirdly, as the author mentioned, increasing pressure for upstream stacks may result in gas leakage, which poses a serious safety risk. As far as I know, the SOFC stack fabricated by Elcogen can only endure a pressure difference of 50 mbar. Are there any actual SOFC systems that employ the serial connection of multiple stacks? Lastly, current large SOFC systems typically integrate multiple stacks within a single hotbox, with stacks connected in parallel in terms of gas flow. I am uncertain whether the stacks in the author's design are all integrated within the same hotbox or if each stack has its own independent hotbox.	requiring diversion to the burner, thus reducing system efficiency. A comparable outcome could also be achieved with a single stack of double capacity, but this would compromise modularity and standardization. We have clarified this rationale in the revised manuscript. Please see page 7, lines 141–151. System practicality and flexibility: We fully agree that the proposed hybrid configuration increases system coupling and therefore reduces local operational flexibility. However, this trade-off is deliberate and consistent with the study's focus on standardization and cost reduction. One of the key challenges in current SOFC or rSOC deployment is that most commercial plants are custom-designed case by case to meet specific power requirements. While this approach provides flexibility, it also leads to extensive redesign effort, including P&ID layout, equipment selection, and pipe sizing, which significantly slows deployment and increases engineering cost. In contrast, the proposed modular design adopts fixed interconnections and standardized modules to enable mass production, simplified troubleshooting, and predictable integration. We acknowledge that higher interdependence between upstream and downstream modules may introduce operational risks. However, the modular structure also facilitates rapid replacement and redundancy. For instance, if an upstream module fails, localized electric heaters or start-up heat exchangers can maintain thermal stability until the affected unit is replaced, without requiring full system shutdown. We have also added future work that will include dynamic operation and control studies to assess these transient scenarios, as this represents an important next step in SOFC scale-up. These points are clarified in the revised manuscript. Please see page 26, lines 632–656; page 14–15, lines 346–362; page 26, Section 2.4, lines 632–656; and page 29, lines 737–744. Pressure considerations: In this study, the maximum allowable pressure drop per stack is set to 100 mbar, which aligns with the tolerance range of SolydEra-type stacks used in existing systems. This value has been cited and discussed in the revised manuscript (page 12, line 298). Regarding the series stack operation, although the complete concept is not yet common, and indeed represents one of the novelties proposed in this study, partial implementations of series-connected stacks already exist. We have clarified this point and added the relevant references in the revised version. Please see page 13, lines 312–317.
--	---	--

		Hotbox arrangement: We appreciate the reviewer’s observation regarding current industrial practice. Large SOFC plants commonly place multiple stacks (and other hot components such as reformers and burners) in a shared hotbox, with low temperature equipment located separately. This can simplify insulation layout and reduce external heat losses, but it also limits independent temperature control for each component and typically requires larger thermal spacing, increasing overall footprint. In the revised manuscript, we clarify that both configurations are feasible:  • Independent hotboxes (module-level enclosures) support plug-in replacement (N+1), shorten hot piping runs (lower heat loss and ΔP), minimize thermal cross-interference, and enable local temperature control. This design is common in the micro-CHP SOFC system, e.g., BlueGen-15 and Sunfire Home750 (please see page 14, line 333). • Shared hotbox designs can reduce enclosure count and interfaces but make it difficult to cool a single module while others remain hot due to the elevated internal air/radiative environment; adequate spacing can also enlarge the system. This design has been validated in the European FP7 project STAGE-SOFC in VTT, Finland (please see page 14, line 333). Our baseline analysis focuses on the modular (independent hotbox) approach because it aligns with the study’s standardization and plug-and-play objectives, thus, in the revised Figure 3 (page 13), we name the grey box as module enclosure boundary. We explicitly note that manufacturers may adopt either arrangement depending on plant size, cost targets, and integration strategy. These clarifications have been incorporated in the revised manuscript. Please see page 14, lines 332–344.
3	I question the necessity and reasonability of employing Monte-Carlo simulation to model the pressure drops of the reformer and burner, given the absence of actual random variables. In Section B2, the assumption that GHSV and geometric parameters follow a normal distribution appears perplexing to me. These parameters are not truly “random” in nature. If	Thank you for the thoughtful comment. We fully agree with you that relying on normally distributed variables for estimating an upper bound of the pressure drops in various components may not be appropriate as their nature is not truly random. In the revision, we employed uniformly distributed samples to accurately estimate conservative values. Further, we opt for the 95th percentile of the distribution as a reasonable probable value. These corrections have been included in the revised manuscript. Please see

	the intention is to perform sensitivity analysis, I would suggest that a uniform distribution seems a more rational choice.	page 8, lines 167-174 and page 34, lines 870 and 873.
4	On page 6, line 124, it states, "Overall, optimum AOG usage within the hybrid system design for scaling up involves two stacks operating in parallel." I'm curious about the rationale behind choosing two as the optimal number of parallel stacks. What led to this specific configuration rather than any other number of stacks operating in parallel?	Thank you for the comment. This comment is similar to the first part of Comment 2. As also mentioned in Comment 2, the two stacks operating in parallel provide the heat required for the subsequent series stack. This balance removes the need for preheating duties upstream of the reformer and stack, while also preventing excessive steam injection into the stack and avoiding unnecessary routing of unconverted fuel to the burner for complete combustion. We have clarified this rationale in the revised manuscript; please see page 7, lines 141–149.
5	Page 7, line 153, why did the stacks operating in series have higher pressure drops? Was it caused by the difference in flowrates?	Thank you for the question. The small differences in stack pressure drop arise from slight variations in flow rate and gas composition across stages. We have added this point, please see page 8, lines 184–186. But in general, we would like to clarify that this does not imply that the system must operate at a single fixed point without flexibility. As shown in Figure 4(a) (page 16), the stack and reformer modules include standardized pipeline connections for fresh fuel and water as well as routing to the burner, which provides operational flexibility and robustness even within a standardized modular architecture. The 50 kW case is presented purely as an illustrative study to ground the theory and aid interpretation. The specific flow rates and pressure drops are provided to give readers a clear sense of scale and to reduce complexity, not to prescribe a unique operating point. We have clarified this in the revised manuscript. Please see page 12, lines 285–291; Figure 3, page 13.
6	On page 2, line 32, it is mentioned that SOFCs have demonstrated minimal degradation. I don't quite agree with this statement. In my view, degradation	Thank you for your question. This is an important and insightful comment. We agree that in the preliminary version, the assumed minimal degradation rate was not sufficiently representative, particularly when compared with PEM fuel cells. The description has been revised accordingly. Please see page 2–3, lines 32–

	and high maintenance cost continue to be a critical issue for the application of SOFCs. The satisfactory lifetimes demonstrated in existing studies are mostly achieved with single cells (even button cells) or short stacks under laboratory conditions. These results may not be directly applicable to real industrial settings.	33. In fact, significant progress has been made in improving SOFC durability in recent years. For example, thousands of BlueGen units operating worldwide, commercialized stack products with mature system designs have demonstrated efficiency degradation rates below 0.2% per 1,000 hours (page 2-3, lines 32-33, reference 13-16). Although further improvement is still needed, the SOFC or rSOC communities are actively advancing degradation mitigation strategies. Given that these technologies are considered essential for future long-term energy storage, particularly in countries that rely heavily on solar energy, we believe continuous progress will make degradation rates increasingly competitive.
7	In Figure 3, you mentioned the heat exchanger module, but I didn't see any heat exchangers in Figure 3 and Figure 4.	Thank you for the comment. We agree that the original figures were confusing. We have revised it. In Figure 3(b), the heat-exchanger module is intended to contain multiple heat-exchangers; it is the most flexible module in the design. We did not specify the exact number because it depends on customer requirements and layout constraints. In our prior work (see page 15, line 370, Ref. 32, 52-56), we used heat-exchanger network (HEN) synthesis/optimization to obtain a fixed layout that minimizes utility demand or the number of heat-exchangers. However, experience from projects such as SWITCH (Ref. 32) shows that industrial partners often prefer to configure connections based on practical considerations (distance, footprint, efficiency, etc.) rather than a fully optimized network. Accordingly, in the revision, we explicitly denote the heat-exchanger module as containing n heat-exchangers, which may include units dedicated to start-up/shutdown or emergency operation (and, where appropriate, electric heaters). By contrast, in Figure 4 the "heater" and "cooler" icons indicate required thermal duties rather than specific components. These clarifications have been incorporated in the revised manuscript. Please see pages 15, lines 364–370; page 13, Figure 3b).
8	In Figure 4, the terms "10kW stack" and "50kW stack" are mentioned. The authors should clarify whether these terms refer to a single stack or to a stack module that consists of multiple stacks integrated within a hotbox.	Thank you for the suggestion. In the proposed design, a total of five stacks is used to deliver 50 kW electrical output: two stacks operate in parallel and three in series, each rated at approximately 10 kW. For two comparative system configurations, the 50-kW capacity can be achieved either by a single 50 kW stack (if available) or by multiple 10 kW stacks connected in parallel, as long as the total electrical

	Additionally, in Figure 3, it is unclear whether the parallel stack modules are located in two independent hotboxes or within the same hotbox.	output equals 50 kW. We have made this point clear in the revised manuscript. Please see page 16, Figure 4 caption. Regarding the hotbox arrangement, it is like Comment 2, part 4; both configurations are feasible. Independent hotboxes (module-level enclosures) allow plug-in replacement (N+1), reduce heat losses and pressure drops by shortening hot piping runs, and enable localized temperature control. In contrast, shared hotbox designs can reduce enclosure count and simplify insulation but restrict individual temperature control and require larger thermal spacing, increasing system footprint. In the revised Figure 3 (page 13), the grey boxes are labeled as module enclosure boundaries. We also note that manufacturers may adopt either configuration depending on plant size, cost, and integration strategy. These clarifications have been incorporated in the revised manuscript. Please see page 14, lines 332–344.
9	On page 15, line 336, it states, “...we have presented a comparative techno-economic analysis of the four scale-up strategies for SOFC system, introduced earlier...”. I found this statement confusing because I couldn't locate any prior introduction to the scale-up strategies. It was only after thoroughly reading through the paper that I discovered the introduction was actually in Section 4.4.1 on page 30.	Thank you for pointing this out, and apologies for the confusion. In the earlier revision, we moved the Methods section to follow the Conclusions; despite our continuity check, this part was still missed. We have corrected it in the revised manuscript and updated the related cross-references. Please see page 19, lines 453–457.
10	Page 26, line 601, you mentioned “more than 10% of the fuel must remain unconverted downstream of the stack”. However, in line 609, you mentioned “AOG primarily contains carbon dioxide, water vapour, and trace amounts of carbon monoxide, hydrogen, and methane.” I think these two sentences are contradictory.	Thank you for the comment. In the revised manuscript, we clarify that 90% refers to the maximum single-pass fuel utilization. Consequently, the anode off-gas from a single stack contains unconverted fuel. In the revised manuscript, the text has been updated accordingly. Please see page 32, lines 796–797 and 805–809.
11	Page 30, line 756, “As illustrated in Figure ???” It appears that there is an issue with the cross-referencing.	Thank you for pointing this out. The issue was caused by a LaTeX formatting error, which has now been corrected. Please see page 36, line 967.

12	Figure 5 illustrates the relationship between LCOE and module size for different system designs. Given that the maximum SOFC module size currently only reaches 100 kW, I believe the standard and prime designs are already sufficient. Moreover, I consider it impractical to construct a 1 GW SOFC module.	Thank you for the suggestion. Before conducting the economic analysis, we discussed and considered the appropriate scope and sizing assumptions. We firstly fixed the overall electricity demand at 1 GW to represent a regional-scale application. The next decision was the maximum size of each component, which is difficult to define a priori. While current SOFC modules are typically around 100 kW, future limits may shift due to engineering advances, stack design, or manufacturing capacity, factors that are uncertain and path-dependent. (Ref 63-67) Our objective is to reveal cost trends across scale-up strategies, not to prescribe today's feasible solution. Therefore, we treat component sizes as variables and do not impose hard feasibility caps that could change with technology evolution. This framing helps identify where economies of scale are most influential and highlights practical questions to the industry, for example, whether larger air blowers or standardized modules could be developed to reduce cost, and whether current limits are driven by engineering constraints or manufacturing capacity. We have clarified this rationale in the revised manuscript. Please see page 19, lines 458–467.
13	In Figure 3, the authors should clarify the meanings of '1X', '2X', and '3X'.	Thanks for the comment. In the revised manuscript, we have updated Figure 3 on page 13 and also explained it in a better way (see page 12, lines 299-303).

Short Highlights to Reviewer 2:

Thank you very much for your thorough review and insightful suggestions. Your comments regarding the interpretation of the results, the conclusions, and the discussion of limitations and future work are all very helpful and have been fully addressed in the revised manuscript.

We would especially like to thank you for your insights on long-term operation and future work. We fully agree with your perspective and have revised the manuscript accordingly. Your remarks also prompted us to reflect critically on our own approach: when advanced concepts are proposed, they must be realized through several intermediate steps rather than a single leap. We view this study as a first, foundational step, and acknowledge that additional steps are required before the concept can be fully validated and implemented. Each of these steps will require substantial data and effort. Our group is already working on these subsequent steps, and we hope to share future outcomes and encourage further joint efforts in this area.

All detailed responses are provided below. Thank you again for your expert feedback and valuable support.

The following Table addresses 1st Revision comments from Reviewer 2.

No.	Comments	Corrections and Responses
1	While the manuscript presents a well-developed system design and techno-economic analysis, it would benefit from a brief discussion of limitations related to long-term operation. In particular, a quantitative or qualitative treatment of component degradation and lifecycle performance, especially for stacks and reformers, could provide a more complete picture of system viability over time. Including a short limitations paragraph addressing these factors, along with the underlying assumptions regarding component scalability and cost behavior, would help contextualize the results and guide future work.	Thank you for the helpful suggestion. We fully agree that our study is based on steady-state analysis and therefore does not capture dynamic behavior such as degradation evolution and long-term operation, robustness under large perturbations, or start-up/shutdown transients. We did not include these aspects because they constitute a substantial and distinct body of work, now underway, and require additional experimental data that were not available at the time of this study. Our intention is that this study provides the foundational system design and TEA framework, and that the next step will be a dynamic optimization and real-time control study under continuous long-time operation, in collaboration with academic and industrial partners to support practical scale-up. To aid readers, we also added a concise synthesis of the cost trends and TRL considerations to summarize the main takeaways. These updates are included in the revision. Please see page 28, Section 3.1, lines 692–720; page 27, lines 674–685. Moreover, before conducting the economic analysis, we discussed and considered the appropriate scope and sizing assumptions. One critical

		decision was the maximum size of each component and its scalability, which is difficult to define a priori. For example, while current SOFC modules are typically around 100 kW, future limits may shift due to engineering advances, stack design, or manufacturing capacity, factors that are uncertain and path-dependent. (Ref 63-67) Our objective is to reveal cost trends across scale-up strategies, not to prescribe today's feasible solution. Therefore, we treat component sizes as variables and do not impose hard feasibility caps that could change with technology evolution. This framing helps identify where economies of scale are most influential and highlights practical questions to the industry, for example, whether larger air blowers or standardized modules could be developed to reduce cost, and whether current limits are driven by engineering constraints or manufacturing capacity. We have also clarified this rationale in the revised manuscript. Please see page 19, lines 458–467.
2	The paper offers a solid comparative analysis of scale-up strategies, showing how different configurations perform across varying system sizes. However, the conclusion could more clearly highlight the trade-offs identified between centralized and modular approaches, particularly the shift in economic viability as system capacity increases. Emphasizing that the optimal strategy depends on scale, fuel price, and component maturity would help reinforce the broader applicability of the findings and underscore the practical considerations involved in SOFC system design.	Thank you for the helpful suggestion. Indeed, our economic results indicate that there is no single optimal strategy; the preferred configuration depends on (i) electricity demand (scale), (ii) operating costs (especially fuel price), and (iii) the TRL/maturity of key components. We emphasize that at larger scales, the proposed hybrid strategy can deliver advantages due to higher efficiency, standardization, and modularity, features that enhance manufacturability and deployment flexibility. In the revised manuscript, we have clarified this on page 27, lines 673–685. In addition, we added a concise synthesis subsection before the Conclusions that explains why standardization and modular design are important for SOFC scale-up. Please see page 26, Section 2.4, lines 632–656.
3	In line 75, the term "AOG" is introduced without definition. It is only defined later (line 608). For clarity, it should be defined at its first occurrence.	Thank you for pointing this out. In the revised manuscript, we added the definition at its first occurrence. Please see page 3, lines 47–48.
4	In Figure 6a, the bar plot showing normalized CAPEX contributions of component modules across scale-up	Thank you for your comment. We agree that when the system size becomes very large and the cost

	strategies at the 1 GW scale includes values of 5%, 6%, 2%, and 2% for strategies C, B, A, and D, respectively. It is unclear which specific equipment or components these values refer to. Please revise the figure or caption for clarity.	decreases, it is difficult to clearly distinguish the data points and labels in the original figure. In the revised manuscript, we have updated the figure and added a zoomed-in inset focusing on the 1 MW and 1 GW cases to improve readability. Please see page 23, Figure 6(a).
5	Line 756 contains a broken or missing reference to a figure. This should be corrected.	Thank you for pointing this out. The issue was caused by a LaTeX formatting error, which has now been corrected. Please see page 36, line 967.

Short Highlights to Reviewer 3:

Thank you very much for your thorough review and insightful suggestions. Your comments are all very helpful and have been fully addressed in the revised version.

We would especially like to thank you for your careful reading of the details and your critical suggestions, in particular, your questions regarding the rationale for the proposed layout (i.e., hybrid design) compared with traditional designs, as well as your insightful remarks on the pressure drop and flexibility of the SOFC system. These points prompted us to think more comprehensively about several aspects that were not fully discussed in the original manuscript, and we have now incorporated these considerations into the revised manuscript. We fully agree with your recommendations and have included the corresponding concepts, extended discussion, and supported references in the revision.

All detailed responses are provided below. Thank you again for your expert feedback and valuable support.

The following Table addresses 1st Revision comments from Reviewer 3.

1	The main claimed innovation—reusing tail gas via an additional SOFC unit arranged in parallel or series—is no longer novel at this stage. There are already numerous publications that have explored similar designs, often with more technical detail and stronger implementation strategies. For instance, this paper simplifies the system-level model by treating the stack pressure drop as a constant. However, such simplification undermines the model's generalizability, as pressure drops vary significantly with stack power capacity and internal flow field geometry. How can this model be scaled? The pressure loss will change entirely,	Thank you for your comments. We understand that several studies have used anode-off (tail) gas recirculation to improve global fuel utilization and avoid external water usage. To overcome the pressure drops, the anode-off gas flow is compressed via a low- or high-temperature blower, and excessive anode-off gas recirculation may offset its primary (i.e., system efficiency) benefits. The proposed hybrid design does not require any blower for anode-off gas recirculation, as anode off-gas from an upstream stage (operating at high pressure) is utilized by a downstream stage (operating at low pressure). Our design uses two stacks in parallel, followed by stacks in series; the rationale behind this choice has been elaborated in response to Comment 2. While multi-stack series configurations are not yet common in commercial systems, the partial concept has been demonstrated at pilot scale. Please see page 13, lines 312–317. From an energetic perspective for SOFC systems with anode-off gas utilization, it is more efficient to operate each stage at a slightly different pressure in the proposed hybrid design than to operate each stage at same pressure with the use of blower to overcome pressure drop for anode-off gas recirculation. The first stage (i.e., parallel stacks), operating at slightly high pressure, can use high pressure of fuel supply and compress water using pump before evaporation. For hybrid design, the power consumption by air
---	---	---

rendering the original conclusions inapplicable.	blowers may increase due to slightly high-pressure operation, but it is compensated by reduced fresh-air flow, as depleted air from parallel stacks is mixed with fresh air and then utilized in series stacks. The proposed hybrid design is innovative and modular, focusing on system efficiency, reduced water usage, and improved thermal performance. This study focuses on the preliminary SOFC system design, to evaluate its techno-economic performance. We have developed (0D) system model in Aspen Plus, which is sufficient for analyzing the conceptual design. The system model includes all practical constraints, including minimum steam-to-carbon ratio, maximum internal reforming, minimum unconverted fuel at the downstream of stack, minimum oxygen concentration in depleted air, and it has been validated against experimental data from several European projects and 50-kW-class demonstration plants. The developed model can be used at different scales, by changing the material and energy flows. The model validation has been presented in our prior publications. Please see page 34, lines 885-887. In the case study (50 kW system), we have used 10 kW stacks that are readily available and can be operated in industrial settings (e.g., maximum allowable temperature and pressure). A scale-up system does not mean that we would need each component with the same capacity. Some of the component modules can be scaled up to the system capacity, whereas others can be scaled out (i.e., multiple modules of a capacity to achieve the system capacity requirement). For example, one MW stack module does not mean a single module; we could use multiple 10- or 100-kW stacks to reach one MW. There is a limitation for the stack design; presently single stack capacity is about 10 kW, while higher capacity is satisfied by module with multiple stacks inside. This means that the pressure drop of each stack/module is fixed and will not be increased after scaling up. The proposed hybrid design requires different pressure operations for different stages. The performance of the proposed hybrid design can be affected by pressure drops for different components. For stacks, reformers, and burners, we have estimated potential pressure drops by varying design and operating variables within feasible ranges. Finally, we have used conservative/maximum pressure drops for different components to confirm better performance of hybrid design compared to traditional design with or without anode-off gas recirculation. In real industrial settings, the pressure drops for different components can be lower than maximum values that would lead to even better performance of hybrid design.
---	---

		Finally, the focus of this study is not on detailed engineering, but rather on performance evaluation and comparison between the proposed hybrid design and existing designs.
2	Another issue with this cascade concept is the considerable increase in system complexity. While not inherently unacceptable, such complex systems demand rigorous system modeling to identify the optimal sizing and configuration—such as how the power ratio between the first-stage and second-stage SOFCs is determined. Is this ratio arbitrarily chosen, based on intuition, or derived through some optimization algorithm? How is self-sustainability of the system ensured, and how does one avoid scenarios where part of the system falls into an energy-deficient state that requires external input?	Thank you for the insightful comments/questions. Regarding the power ratio between the stack modules connected in parallel and those connected in series; we appreciate the opportunity to clarify the rationale behind this design choice. The main motivation is to maximize the utilization of high-temperature heat through direct mixing. Our goal is to harness (thermal and chemical) energy of the AOG flow not only for unconverted fuel recovery but also for preheating the inlet flows to the reformer and stack inlets, thereby minimizing unconverted fuel or heat loss to the burner. To achieve thermal balance, the total flow from two parallel stacks ($2 \times X$ kW) represents the minimum requirement for providing the necessary preheating duty to the subsequent reformer and stack. Further, this configuration can optimally supply steam from an upstream stage to the subsequent downstream stage. While alternative configurations, such as three stacks connected in parallel, are technically feasible, they would lead to heat imbalance, with surplus flow requiring diversion to the burner, thus reducing system efficiency. A comparable outcome could also be achieved with a single stack of double capacity, but this would compromise modularity and standardization. We have clarified this rationale in the revised manuscript. Please see page 7, lines 141–149. Regarding self-sustainability, we acknowledge that higher interdependence between upstream and downstream modules may introduce operational risks. However, the modular structure also facilitates rapid replacement and redundancy. For instance, if an upstream module fails, localized electric heaters or start-up heat exchangers can maintain thermal stability until the affected unit is replaced, without requiring full system shutdown, thanks to the modular approach. We have also added future work that will include dynamic operation and control studies to assess these transient scenarios, as this represents an important next step in SOFC scale-up. Moreover, we agree that the proposed hybrid configuration increases system coupling and therefore increases the system complexity and reduces local operational flexibility. However, this trade-off is deliberate and consistent with the study's focus on standardization and cost reduction. One of the key

		challenges in current SOFC or rSOC deployment is that most commercial plants are custom-designed case-by-case to meet specific power requirements. While this approach provides flexibility, it also leads to extensive redesign effort, including P&ID layout, equipment selection, and pipe sizing, which significantly slows deployment and increases engineering cost. In contrast, the proposed modular design adopts fixed interconnections and standardized modules to enable mass production, simplified troubleshooting, and predictable integration. Overall, we agree that additional work is needed to convert this concept into real-world deployment. Even so, we regard this study as an important first step that offers foundational insights and design guidance for industry partners. Advancing a large-scale SOFC system from low to high TRL is a staged process; each stage involves significant effort, not a single ‘one-step’ transition. These points are clarified in the revised manuscript. Please see page 26, lines 632–656; page 14–15, lines 346–362; page 26, Section 2.4, lines 632–656; and page 29, lines 737–744.
3	The economic model is overly optimistic. The final projected electricity cost is not the result of system optimization but rather a consequence of assuming reduced CAPEX due to large-scale manufacturing. In reality, SOFC demand is unlikely to reach such scales.	Thank you for the suggestion. Before conducting the economic analysis, we discussed and considered the appropriate scope and sizing assumptions. We firstly fixed the overall electricity demand at 1 GW to represent a regional-scale application. The next decision was the maximum size of each component, which is difficult to define a priori. While current SOFC modules are typically around 100 kW, future limits may shift due to engineering advances, stack design, or manufacturing capacity, factors that are uncertain and path-dependent (Ref 63-67). Our objective is to reveal cost trends across scale-up strategies, not to prescribe today’s feasible solution. Therefore, we treat component sizes as variables and do not impose hard feasibility caps that could change with technology evolution. This framing helps identify where economies of scale are most influential and highlights practical questions for industry, for example, whether larger air blowers or standardized modules could be developed to reduce cost, and whether current limits are driven by engineering constraints or manufacturing capacity. We have clarified this rationale in the revised manuscript. Please see page 19, lines 458–467.

4	The cascade design is already well explored in the literature but the dynamic operation with more modeling details can be a future direction, or the experimental optimization of such system as it's really challenge to achieve self-sustainability.	Thank you for the comment. 1. Regarding the heat cascade point, the straightforward and most deployed layout today is no AOG, no COG (NCNA) system, largely because the configuration is simpler and easier to control. However, NCNA faces several limitations. First, even small efficiency gains matter, especially when SOFC systems are paired with reversible operation (i.e., SOEC) for long-duration storage, as small percentage improvements compound over time. For this reason, indeed, several European demonstration projects (e.g., CH2P, SWITCH, Nautilus, BLAZE, AMON) have implemented either AOG or COG recirculation (all related with heat cascade topic – but only recirculate back to the original stack). Most designs adopt warm AOG recirculation: the AOG is cooled to ~280 °C and compressed with a blower (commercial blowers are typically limited to this temperature), then compressed flow is injected upstream of the reformer and reheated to the reformer inlet temperature. AOG recirculation can increase electrical efficiency and exploit the steam content of the off-gas, thereby reducing fresh-fuel demand and the heat otherwise required to generate external steam. Less fuel lowers cost; less required heat frees more waste heat for valorization; and reduced external water demand allows for a smaller water-purification unit, which can be costly in many regions (on the order of 3 €/m³ water, depending on quality). That said, our analysis also shows that the commercial AOG blower can offset or even negate these efficiency benefits, an important practical limitation. This is precisely the issue our work addresses: how to retain the benefits of AOG recirculation while avoiding dependence on a blower. Based on the literature review, we believe our proposed strategy is able to address this issue while maintaining the recirculation benefits. In the revised manuscript, we have clarified these points and cited supported references. Please see page 3, lines 40-53; Page 26, Section 2.4 for our clarifications. 2. Regarding the dynamic optimization, we fully agree that our study is based on steady-state analysis and therefore does not capture dynamic behavior such as degradation evolution and long-term operation, robustness under large perturbations, or start-up/shutdown transients. We did not include these aspects because they constitute a substantial and distinct body of work, now underway, and require additional experimental data that were not available at the time of this study. Our intention is that this study provides the foundational system design and TEA framework, and that the next step will be a dynamic optimization and real-time control study under continuous long-time operation, in collaboration with academic and industrial partners
---	---	---

		to support practical scale-up. To aid readers, we also added a concise synthesis of the cost trends and TRL considerations to summarize the main takeaways. These updates are included in the revised manuscript. Please see page 28, Section 3.1; page 27, lines 671–683. 3. Finally, regarding self-sustainability, it is like Comment 2. The standardized architecture proposed here imposes tighter inter-module interfaces and reduces component-level customization. Compared with conventional case-by-case designs, however, it lowers footprint and integration complexity and, in return, provides system-level flexibility: capacity can be scaled to the required size with minimal re-engineering and predictable BoP. In other words, the approach shifts flexibility from bespoke component choices to deployability for users/operators, which we consider essential for scale-up. We acknowledge that additional work is critically needed on emergency operation, troubleshooting, start-up/shutdown, and long-term operation. Each of these is a substantial but necessary step toward the future large-scale deployment, and this study provides the foundation for those efforts. These points are clarified in the revised manuscript. Please see page 26, lines 632–656; page 14–15, lines 346–362; page 26, Section 2.4, lines 632–656; and page 29, lines 737–744.
5	A simple review would reveal numerous works offering more nuanced modeling or experimental verification. On top of this, the modeling details in the present work are insufficient, and the model’s applicability to larger systems is questionable. It may only be valid for sub-kilowatt stacks, in which case, stack-level simplifications of this kind are not acceptable.	Thank you for the comment. Our system model has been validated against experimental data from several European projects and 50-kW-class demonstration plants. The validation procedure has been documented in our prior publications; to avoid redundancy we do not repeat it here, but we have explicitly cited our previous studies in the revised manuscript. See references 52, 56, 58, 77, 82-84. Accordingly, we do not fully agree that the model is simplified as it is supported by multiple operation data. We acknowledge, however, that most available validations are at the tens-of-kilowatts scale. This raises a fair question about when whole-plant validation becomes critical at larger capacities. In practice, full experimental validation of several-hundred-kilowatt plants is rarely feasible at low TRL and is typically confined to a few large industrial programs. Further, proceeding directly to purchase and test only very large bespoke units would be risky and inefficient. Our approach

		therefore follows a component/module validation strategy: we use experimental data for critical components across small-to-medium scales and assemble the system from standardized modules. In prior studies, spanning a few kilowatts up to ~100 kW, we did not observe scale effects that invalidate component performance within the operating envelopes used here. We agree that, at a later stage (higher TRL), integrated whole-plant validation will be necessary; this is beyond the scope of the present work. We have included it in the “Limitations and Future Work” section (page 28) and clarified the component-level validation in the Methods. Please see page 34, lines 883–897, in the revised manuscript.
6	The economic model is too idealized, producing a highly attractive electricity price figure that is unrealistic—even coal-fired or gas turbine power plants cannot deliver power at \$0.155/kWh in current market conditions.	Thank you for your thoughtful feedback. An electricity price as low as \$0.155/kWh at very large scale is realistic. According to the International Energy Agency, the LCOE for coal-fired power plants remains below \$0.10/kWh, and many gas-turbine configurations are also under this threshold (sources: https://www.iea.org/data-and-statistics/charts/lcoe-and-value-adjusted-lcoe-for-solar-pv-plus-battery-storage-coal-and-natural-gas-in-selected-regions-in-the-stated-policies-scenario-2022-2030; https://www.iea.org/data-and-statistics/charts/lcoe-for-gas-with-and-without-ccus-for-various-carbon-prices). In some countries (e.g., China and India), the LCOE for conventional thermal generation is even lower. That said, achieving \$0.155/kWh for SOFCs requires very large plant size, as indicated by our results. We view this value as a potential outcome at scale, not a universal near-term target. Overall, there is no single optimal low-cost strategy. The preferred configuration depends on (i) electricity demand (scale), (ii) operating costs (especially fuel price), and (iii) the TRL/maturity of key components. At larger scales, the proposed hybrid strategy can offer advantages due to higher efficiency, standardization, and modularity, features that enhance manufacturability and deployment flexibility. In general, our economic analysis focuses on cost trends, not on a single headline value. Accordingly, we treat component sizes as variables and do not impose hard feasibility caps that may change with technology evolution. This framing identifies where economies of scale matter and highlights practical questions for industry, for example, whether larger air blowers or standardized modules could be developed to reduce cost, and whether current limits are driven by engineering constraints or manufacturing

		capacity. We also include sensitivity analyses on key parameters (fuel price, component costs/TRL, plant size) to reflect uncertainty. We have clarified these points in the revised manuscript. Please see page 21, line 501; page 27, lines 671–683; page 19, lines 458–467.
--	--	--

Short Highlights to Reviewer 4:

Thank you very much for your thorough review and insightful suggestions. Your comments regarding the future work and reference citation are very helpful and have been fully addressed in the revised manuscript.

All detailed responses are provided below. Thank you again for your expert feedback and valuable support.

The following Table addresses 1st Revision comments from Reviewer 4.

1	At the beginning of the article, they point out the reversibility of SOC technology, but ultimately focus more on the SOFC fuel cell. However, it would be more interesting to see how the design affects the SOEC. Both may require a different setup at the system level. In this case, the rSOC character would be ineffective. The authors do not address industrial interconnection sufficiently at the end – Bloom Energy is mentioned.	Thank you for the helpful comment. Our overarching goal is to propose a scale-up strategy for rSOC systems. When initiating this work, we recognized the value of treating rSOC as two coupled subsystems and performing detailed studies for each. We began with the SOFC side because it is generally more complex and involves a larger set of design variables and operating constraints; this study therefore focuses on laying that foundation. We fully agree that SOEC must be addressed to discuss rSOC scale-up comprehensively. To maintain a clear scope and avoid diluting the message, we deferred the SOEC analysis to a dedicated follow-on study. That work is ongoing and builds directly on the modular architecture developed here for SOFCs. In the revised manuscript, these points and the planned SOEC work are highlighted. Please see page 28, lines 712–716.
2	Overall, the article is well-written. A weakness becomes apparent in the literature review: too few citations (41) for such an "old" and well-known research area. Furthermore, the article lacks inclusion of publications from important European and American companies, universities, and research institutions.	Thank you for your valuable input. While we believe the overall concept in our study is relatively novel, and that few prior studies cover the entire scope end-to-end, we agree that parts of the concept and several claims can be supported from multiple perspectives. In the revised manuscript, we have expanded the literature base to 81 references, including from prominent companies, universities, and research institutions:  • Companies: SolydEra, New Enerday, Sunfire, Bloom Energy, Noon Energy, Dynelectro, among others.

		• Universities and Research Institutions: Technical University of Denmark, University of Tehran, Imperial College London, VTT Technical Research Centre of Finland, Aalborg University, Harvard University, Tsinghua University, Peking University, and others.
--	--	---